# FinTSBridge: A New Evaluation Suite for Real-world Financial Prediction with Advanced Time Series Models

**Yanlong Wang**[1,2]    **Jian Xu**[1]    **Tiantian Gao**[1]    **Hongkang Zhang**[1]    **Shao-Lun Huang**[1]
**Danny Dongning Sun**[2*]  **Xiao-Ping Zhang**[1*]

[1]Tsinghua University  [2]Peng Cheng Laboratory

## Abstract

Despite the growing attention to time series forecasting in recent years, many studies have proposed various solutions to address the challenges encountered in time series prediction, aiming to improve forecasting performance. However, the strategic interactions inherent in financial markets make it challenging to effectively apply time series forecasting models to asset pricing. There is still a need for a bridge to connect cutting-edge time series forecasting models with financial asset pricing. To bridge this gap, we have undertaken the following efforts: 1) We constructed three datasets from the financial domain; 2) We selected over ten time series forecasting models from recent studies and validated their performance in financial time series; 3) We developed new metrics, msIC and msIR, to quantify specific aspects of time series correlation captured by the models; 4) Task designs tailored to our financial data collections enabled rigorous evaluation of models' operational effectiveness in financial decision-making contexts. Our findings suggest that time series forecasting models, through domain-specific processing and evaluation, exhibit effective performance across diverse asset classes and sampling frequencies. We hope the developed new evaluation suite, FinTSBridge, can provide valuable insights into the effectiveness and robustness of advanced forecasting models for financial time series.

## 1 Introduction

Time series forecasting has become increasingly crucial in the financial domain, playing a critical role in decision-making processes related to asset pricing, risk management, and algorithmic trading [42, 28, 35]. Accurate forecasts are essential for tasks such as stock index prediction, option pricing, and modeling cryptocurrency volatility, as they help optimize investment strategies and mitigate market risks [26, 17]. However, the non-stationary nature of financial markets-shaped by factors like geopolitical events and investor sentiment-adds significant complexity to these challenges [58]. Despite advancements in machine learning for time series analysis, transforming state-of-the-art models into actionable financial insights remains an ongoing obstacle [15]. For example, while models may achieve low mean squared error (MSE) on synthetic datasets, their performance often deteriorates in real-world conditions due to unaccounted market dynamics, such as price-volume interactions and multi-scale volatility.

Moreover, many studies focus on time series datasets with simplified statistical properties, such as electricity consumption or traffic flow, which typically exhibit strong stationarity and periodicity [13]. Even commonly used financial datasets, such as historical exchange data, suffer from key limitations: daily resolution obscures intraday fluctuations, and limited variable coverage (e.g., the absence of derivatives metrics) fails to capture the full spectrum of multi-scale market interactions

---

*Corresponding authors.

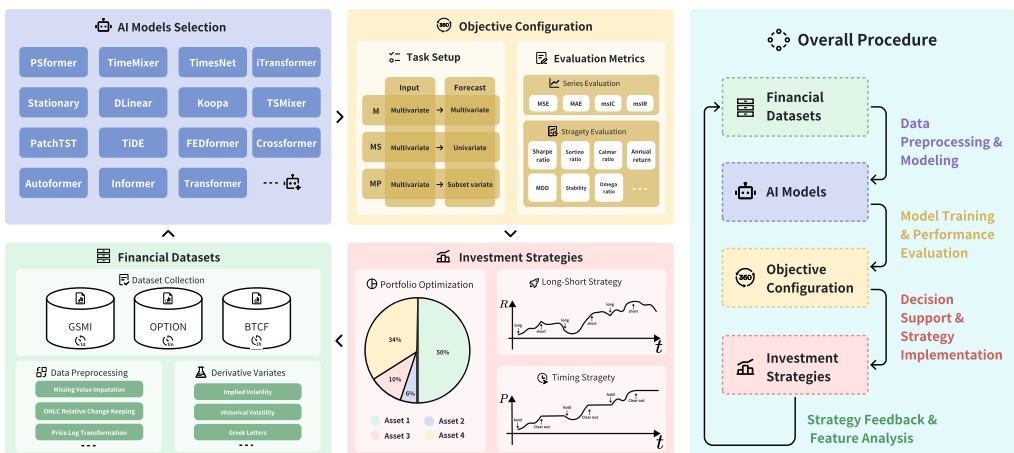

Figure 1: The overall pipeline of FinTSBridge. First, three financial datasets are constructed and corresponding preprocessing is carried out. Then, a wide range of time series models are utilized for training. Through extensive task settings and comprehensive evaluations, the performance of the models is verified. Finally, investment strategies are formulated in real-world financial scenarios and the performance of these strategies is evaluated.

[4]. Compounding these issues, traditional evaluation metrics like MSE and mean absolute error (MAE) prioritize point-wise accuracy while neglecting temporal correlations, a critical aspect in financial forecasting. For instance, our empirical results show that a naive model that predicts the last observed value can achieve competitive MSE but exhibits near-zero correlation with true market trends, making it ineffective for practical trading strategies. These shortcomings underscore the need for financial-specific benchmarks that reflect the complexity of real-world markets, along with evaluation frameworks that assess both predictive accuracy and economic utility.

To address these gaps, we introduce FinTSBridge, a comprehensive framework that connects state-of-the-art time series models with real-world financial time series and applications. As depicted in Figure 1, our process begins with the pre-processing of financial data. We have developed a specialized method tailored to financial time series that enhances stationarity while preserving the interrelationships between variables, facilitating more effective model training. The data is then fed into the modeling phase, where we apply various AI-driven time series models to tackle diverse forecasting tasks. We design multi-perspective forecasting tasks and evaluate model performance using both traditional and novel metrics, complemented by financial metrics for assessing strategy performance. Depending on practical requirements, we assess models not only for predictive accuracy but also for their potential to support investment strategies through strategy simulation. This process aims to provide actionable decision support while deepening our understanding of financial data and processing techniques.

By introducing new datasets, evaluation metrics, and forecasting tasks, FinTSBridge seeks to bridge the gap between advanced time series models and the complex challenges encountered in financial markets. Our work emphasizes the importance of a correlation-aware approach to financial time series forecasting, ultimately advancing the field of AI-driven finance. Our key contributions can be summarized as follows:

- Three curated financial datasets are constructed to reflect diverse market dynamics:
  - Global Stock Market Indices (GSMI): 20 indices (2005–2024), capturing cross-market dependencies and volatility regimes.
  - High-Frequency Option Metrics (OPTION): Minute-level implied volatility and Greeks for CSI 300 ETF options (2024), modeling intraday market microstructure.
  - Bitcoin Futures-Spot Dynamics (BTCF): Hourly price-volume sequences (2020–2024), capturing crypto market lead-lag effects.

- The performance validation of over ten leading time series forecasting models from recent research works on financial time series is conducted to showcase their practical viability in this domain.

- Two new evaluation metrics, msIC (mean sequential correlation) and msIR (correlation stability ratio), are developed as a complement to traditional metrics like MSE and MAE, to better capture temporal correlation and robustness under non-stationarity.

- Financial-specific tasks over these datasets, such as index portfolio optimization, Timing trading and BTC futures long&short strategies, are further designed to assess the practical performance and application potential of forecasting models.

## 2 Related Work

### 2.1 Time-series Forecasting Models

Recent advances in time-series forecasting have focused on enhancing model capabilities through multi-scale decomposition, attention mechanism optimization, lightweight architectures, and representation learning innovations. Decomposition-based methods remain foundational for handling non-stationary signals. For example, Autoformer [53] pioneers autocorrelation-driven periodicity detection, replacing traditional moving averages with adaptive decomposition, while FEDformer [60] leverages Fourier-Wavelet hybrid spectral analysis to achieve multi-resolution frequency decomposition. Building on this, Non-stationary Transformers [31] introduce dynamic stationarization modules that learn to normalize non-stationary inputs before applying attention mechanisms, significantly improving robustness to distribution shifts. These approaches highlight the synergy between frequency-domain analysis and temporal modeling.

To address computational bottlenecks in Transformers, researchers have developed sparse attention variants: Informer [59] reduces quadratic complexity via probabilistic sparse attention with KL-divergence-based token selection, while Crossformer [57] designs hierarchical cross-resolution attention to capture dependencies across temporal scales. Koopa [30] bypasses attention entirely, proposing time-varying Koopman dynamic systems that model latent state evolution through linear operators. Concurrently, lightweight architectures challenge conventional wisdom-DLinear [56] demonstrates that decoupled linear projections for trend and residual components can outperform complex models, while TSMixer [12] and TimeMixer [46] adopt pure MLP-based architectures and hybrid frequency-time operators, respectively, to balance accuracy and computational cost.

Representation learning breakthroughs further expand modeling capabilities. TimesNet [52] transforms 1D sequences into 2D temporal matrices via period-phase folding, enabling 2D convolutions to capture intra-period and inter-period patterns simultaneously. PatchTST [38] introduces channel-independent patching inspired by vision transformers, learning localized temporal embeddings through overlapping segments. iTransformer [29] inverts the traditional architecture by treating variates as tokens and time points as features, enhancing multivariate dependency modeling. MICN [45] employs multi-scale dilated convolutional pyramids to extract local periodic features hierarchically. Complementary techniques like RevIN [23] address distribution shifts via bidirectional instance normalization, and TiDE [14] integrates temporal encoding with dense residual connections for efficient long-horizon forecasting. These innovations collectively advance three core principles: 1) Hybridization of frequency-temporal analysis, 2) Strategic simplification guided by signal processing theory, and 3) Systematic robustness enhancement through normalization and distribution alignment.

### 2.2 Financial task-related Studies

Recent advances in financial time series forecasting have increasingly adopted multi-horizon prediction frameworks to capture evolving market dynamics. Unlike traditional approaches that focus on single-step forecasting (e.g., predicting next-step prices or up-down trends as in [16, 1, 24, 40, 27]), recent time series methods like [29, 52, 30] employ sequence-to-sequence architectures to predict multi-step trajectories of exchange time series, which helps to understand future temporal dynamics but also introduces challenges for prediction. Although large models and Reinforcement Learning methods are becoming increasingly popular in financial time series forecasting [39, 27, 61], the potential of small models and supervised learning approaches in financial time series forecasting has

not been fully explored. It is important to build a bridge between cutting-edge models and financial time series tasks.

Although end-to-end forecasting methods have become increasingly popular in the time series domain, introducing covariates as part of the prediction in financial datasets is necessary. These covariates provide time-varying information that the time series itself does not possess, posing challenges for models to capture inter-variable information. Additionally, different time series scales can lead to varying prediction performance and dependence on covariates. In financial time series, phenomena that are difficult to predict or do not appear in low-frequency time series may exhibit predictability in high-frequency time series. This makes it valuable to construct a financial time series dataset that covers different frequencies, financial instruments, and variable types, as it facilitates a comprehensive evaluation of financial time series forecasting capabilities.

## 2.3 The Predictability of Asset Prices

Stock price prediction is indeed challenging, but it is not entirely impossible. This is evidenced by the long-standing work of quantitative hedge fund institutions, which continuously mine predictive indicators and patterns from historical datasets to cope with the ever-changing market environment, achieving excess returns. Moreover, some studies based on the Efficient Market Hypothesis (EMH) suggest that the stock market is not necessarily in the semi-strong form or strong-form efficiency as previously thought, which would prevent future price movements from being forecasted based on available historical information [36, 34]. On the contrary, many markets often fall between semi-strong and weak-form efficiency [33], and [18] has validated the momentum effect in the stock market, where stocks that performed well in the past may continue to perform well in the future, thereby challenging the weak-form efficient market hypothesis. [19] discusses the explanatory power of multi-factor models for asset pricing anomalies, indirectly indicating the level of market efficiency. Additionally, advancements in technology have significantly enhanced data processing capabilities, leading to improvements in the performance of predictive models. This development brings data processing power and predictive performance that were previously unavailable in models based solely on historical data [10, 25, 22, 20, 5].

## 3 Dataset Curation

Current research on long-term time series forecasting predominantly focuses on eight mainstream datasets [52, 12, 29, 30, 56, 38]. Among these, five are related to electricity, while the remaining three pertain to weather, traffic, and exchange rates. Although exchange rate data falls under the financial domain, it is often overlooked compared to the other seven datasets or replaced by the ILI dataset from the illness area, due to the non-stationary and non-periodic characteristics of exchange rate data [32], which makes it more challenging to predict than other time series data that exhibit significant periodicity and stationarity.

While some state-of-the-art time series models have demonstrated strong long-term forecasting capabilities on mainstream datasets, they lack the robustness needed to tackle the complexities of real-world time series problems [41, 6, 8, 7]. To better investigate these real-world challenges, we propose three financial time series datasets.

## 3.1 Data Sources

We have constructed three financial time series datasets: GSMI, OPTION, and BTCF, each representing different subfields of finance. The GSMI dataset includes 20 major indices from global stock markets, recording daily price and trading volume data for these indices over nearly 20 years, from 2005 to 2024. The OPTION dataset includes the CSI 300ETF options from the Chinese financial market, with variables related to risk for both call and put options. The BTCF dataset contains hourly frequency data for Bitcoin spot and perpetual contracts, helping to understand the spot-contract lag relationship and facilitate long-short trading strategies [11, 37]. In Table 3, we present the statistical properties of these three financial time series datasets with more details.

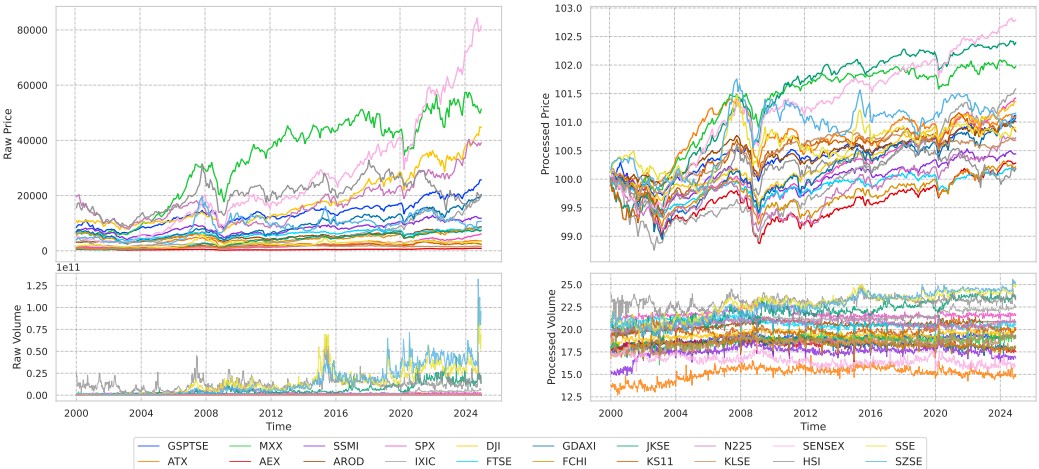

Figure 2: Comparison of Price Series of 20 Major Global Stock Markets. This image compares the close price series of 20 major global stock markets before and after standardization. By applying a logarithmic transformation and normalizing to a base of 100, the time series becomes more stable, which benefits model training and allows for easy reversal to the original series.

## 3.2 Data Preprocessing Methods

Due to the significant differences in the magnitude of value changes across many variables in the raw data, appropriate data preprocessing is necessary. However, there is no unified and consistent solution for preprocessing asset price sequences, and the preprocessing methods need to be constructed based on the specific task and requirements. For the GSMI and BTCF datasets, since their frequencies are daily and hourly, respectively, we use high-open-low-close prices to assist in capturing the data's information. To achieve this, we consider applying logarithmic transformations while preserving the relative change patterns between these variables.

Suppose the asset close price series is given by

$$P_{0,t}^c = \{p_0^c, p_1^c, ..., p_i^c, ..., p_t^c\}, \tag{1}$$

where $p_i^c$ represents the close price at the $i$-th time step, and $i \in \{0, 1, \ldots, t\}$. Let close price change at the $i$-th time step as

$$R_i^c = \frac{p_i^c}{p_{i-1}^c}, \tag{2}$$

then $p_i^c = p_{i-1}^c \cdot R_i^c$, which leads to $P_{0,t}^c = \{p_0^c, p_0^c \cdot R_1^c, ..., p_{i-1}^c \cdot R_i^c, ..., p_{t-1}^c \cdot R_t^c\}$. Therefore, after the logarithmic transformation, we construct

$$ln(P_{0,t}^c) = ln(p_0^c) + \{0, ln(R_1^c), \ldots, ln(\prod_{i=1}^{t} R_i^c)\}. \tag{3}$$

Then, subtracting the initial $ln(p_0^c)$ leads to

$$ln(\frac{P_{0,t}^c}{p_0^c}) = \{0, ln(R_1^c), \ldots, \sum_{i=1}^{t} ln(R_i^c)\} \tag{4}$$

At this point, the constructed log price series can be viewed as the cumulative sum of price change, and the transformed sequence depends only on the price change from the previous state, showing a additivity property.

Similarly, for the asset high price series given by $P_{0,t}^h = \{p_0^h, p_1^h, ..., p_i^h, ..., p_t^h\}$, where $p_i^h$ represents the high price at the $i$-th time step. Let the high price relative to last close price change at the $i$th time step as

$$R_i^h = \frac{p_i^h}{p_{i-1}^c}, \tag{5}$$

then $p_i^h = p_{i-1}^c \cdot R_i^h$. Therefore, the final logarithmic transformation for high price series as

$$ln(\frac{P_{0,t}^h}{p_0^c}) = \{0, ln(R_1^h), \dots, \sum_{i=1}^t ln(R_i^h)\} \tag{6}$$

This not only allows the high price series to exhibit additivity properties but also retains the relative relationship between the high and close prices, as their difference can be expressed as

$$\Delta(P^h, P^c) = ln(\frac{P_{0,t}^h}{p_0^c}) - ln(\frac{P_{0,t}^c}{p_0^c}) = \{0, ln(\frac{R_1^h}{R_1^c}), \dots, ln(\prod_{i=1}^t \frac{R_i^h}{R_i^c})\}, \tag{7}$$

where we have

$$\frac{R_i^h}{R_i^c} = \frac{p_i^h}{p_i^c}, \tag{8}$$

which is uniquely determined by the price at the $i$-th step. Additionally, we add a constant term of 100 to the transformed sequence to anchor the baseline for cumulative changes and prevent negative values in the log price series. The final preprocessed price becomes

$$Z_{0,t} = ln(\frac{P_{0,t}}{p_0^c}) + 100, \tag{9}$$

where $Z_{0,t}$ represents the transformed price series, and $P_{0,t}$ can be any of the open, high, low, or close price series.

For the trading volume series $V_{0,t} = \{v_0, v_1, v_2, \dots, v_i, \dots, v_t\}$, the logarithmic transformation method is

$$Z_{0,t}^v = ln(V_{0,t} + 1), \tag{10}$$

which helps avoid errors in the logarithmic calculations when the trading volume is zero.

### 3.3 Visualization of Preprocessing

In Figure 2, we provide a comparison of the 20 index closing price and trading volume sequences from the GSMI dataset before and after preprocessing. Prior to preprocessing, these index price sequences exhibit larger fluctuations and varying magnitudes, making cross-sectional comparisons challenging. The temporal variations in the trading volume sequences are even more unstable, and comparing trading volumes across indices is particularly difficult. After preprocessing, both the price and trading volume sequences are maintained within the same magnitude range and price series are anchored to a unified initial baseline, enhancing the comparability and consistency of cumulative changes.

We also provide comprehensive analyses of the changes in the Volume-Price series before and after preprocessing to show its efficacy in aligning data scales and preserving key patterns across variables. For the GSMI dataset, the transformations of each index price sequence can refer to Appendix A.1.2 and Appendix A.1.3 illustrates how the relative properties between variables are preserved. The comparison of BTC spot candlestick charts for the BTCF dataset before and after preprocessing is presented in Appendix A.3.2.

## 4 New Evaluation Metrics

While mainstream time series forecasting works in top conferences often adopt error-based methods, such as MSE and MAE, as evaluation metrics for model predictions, these metrics face significant challenges when applied to financial time series. A simple example is that a Naive model as in Table 13, which directly uses the last observed value of the input sequence as the forecast, achieves remarkably low prediction errors on the Exchange dataset (we provide detailed analysis in Appendix B.5). This raises critical questions about the evaluation of financial time series forecasting, suggesting that correlation-based metrics should be introduced alongside prediction errors, as they are crucial for real-world financial applications (more details at Appendix D). Although traditional Information Coefficient and Information Ratio metrics [43, 21] can measure temporal correlations

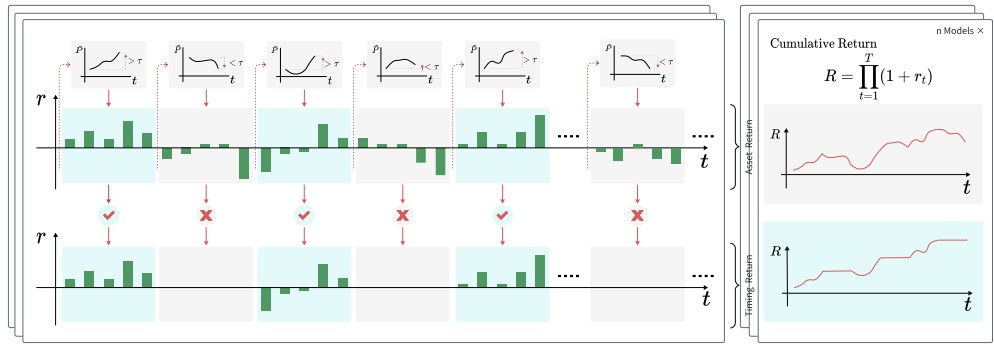

Figure 3: Timing Process. At each trading stage, the magnitude of the price change predicted by the model is compared with the threshold at that time. If it is greater than the threshold, a buying strategy is executed, and the subsequent returns are obtained until the next trade. If it is less than the threshold, cash is held, so the return for this stage is 0 until the next trade. The cumulative return curve is obtained by multiplying the return rate sequences of these trades.

in single-step univariate forecasting, they fail to assess multi-variable multi-step predictions. To address this limitation, we propose multi-step IC and multi-step IR, abbreviated as msIC and msIR, respectively.

msIC measures the correlation coefficient between the true and predicted values of the forecast time series over the prediction horizon. Specifically, for input data consisting of B samples, represented as $X \in \mathbb{R}^{B \times L \times C}$, where $B$ is the number of samples, $L$ is the sequence length, and $C$ is the number of variables, after mapping through a neural network $f$, we obtain $\hat{Y} = f(X; \theta)$, where $Y, \hat{Y} \in \mathbb{R}^{B \times F \times C}$, with $F$ being the forecast horizon. msIC is used to measure the temporal correlation between the predicted time series $\hat{Y}$ and the true time series $Y$. Specifically, we compute the rank correlation coefficient for each sample and each variable along the time dimension, and then average over the B and C dimensions to get the final value.

For the $i$-th sample and $j$-th variable, the rank correlation coefficient for the predicted time series is given by

$$\rho_{Y_{i,j}, \hat{Y}_{i,j}} = \frac{\text{Cov}(Y_{i,j}, \hat{Y}_{i,j})}{\sigma_{Y_{i,j}} \sigma_{\hat{Y}_{i,j}}}, \tag{11}$$

where $Y_{i,j}$ and $\hat{Y}_{i,j}$ are the time series for the $i$-th sample and $j$-th variable. Then, $msIC$ is represented as

$$msIC = \frac{1}{B \times C} \sum_i^B \sum_j^C \rho_{Y_{i,j}, \hat{Y}_{i,j}}. \tag{12}$$

Although msIC effectively reflects the correlation between the predicted and true time series, it does not account for the temporal fluctuations of this correlation due to the time-varying distribution of the time series. Therefore, we also construct msIR to capture this aspect. Specifically, for the $i$-th sample, the cross-channel correlation can be expressed as

$$msIC_i = \frac{1}{C} \sum_j^C \rho_{Y_{i,j}, \hat{Y}_{i,j}}, \tag{13}$$

and $\{msIC_1, msIC_2, ..., msIC_B\} \in \mathbb{R}^B$ maintains strict chronological order. The standard deviation of these values is given by

$$\sigma = \sqrt{\frac{1}{B} \sum_{i=1}^B (msIC_i - msIC)^2}, \tag{14}$$

where $msIC = \frac{1}{B} \sum_i^B msIC_i$. Finally, $msIR$ is calculated as

$$msIR = \frac{msIC}{\sqrt{\frac{1}{B} \sum_{i=1}^B (msIC_i - msIC)^2}}. \tag{15}$$

Table 1: Main Results. All the results are averaged from 4 different prediction lengths $H \in \{5, 21, 63, 126\}$ and 5 runs with different seed. A lower MSE or MAE indicates lower prediction error, while a higher msIC or msIR indicates higher prediction correlation.

| | Metric | PSformer | TimeMixer | Koopa | iTransformer | TiDE | PatchTST | DLinear | Stationary | TSMixer | TimesNet | FEDformer | Autoformer | Crossformer | Transformer | Informer | Naive |
|---|---|---|---|---|---|---|---|---|---|---|---|---|---|---|---|---|---|
| GSMI | MSE↓ | 0.112 | 0.124 | 0.124 | 0.126 | 0.131 | 0.124 | 0.136 | 0.187 | 0.220 | 0.230 | 0.245 | 0.281 | 0.713 | 1.055 | 1.154 | 0.149 |
| | MAE↓ | 0.191 | 0.205 | 0.205 | 0.209 | 0.210 | 0.200 | 0.230 | 0.286 | 0.329 | 0.332 | 0.364 | 0.383 | 0.550 | 0.789 | 0.807 | 0.194 |
| | msIC↑ | 0.068 | 0.020 | 0.021 | 0.021 | 0.028 | 0.012 | -0.036 | 0.042 | -0.021 | 0.005 | 0.052 | 0.039 | 0.004 | 0.062 | 0.024 | 0.000 |
| | msIR↑ | 0.224 | 0.099 | 0.081 | 0.090 | 0.138 | 0.106 | 0.018 | 0.137 | -0.071 | 0.010 | 0.215 | 0.141 | 0.050 | 0.193 | 0.060 | -0.001 |
| OPTION | MSE↓ | 0.251 | 0.255 | 0.267 | 0.266 | 0.255 | 0.256 | 0.256 | 0.395 | 0.639 | 0.446 | 0.594 | 0.617 | 0.311 | 1.344 | 2.081 | 0.370 |
| | MAE↓ | 0.229 | 0.233 | 0.244 | 0.240 | 0.229 | 0.234 | 0.238 | 0.316 | 0.531 | 0.337 | 0.482 | 0.481 | 0.322 | 0.877 | 1.125 | 0.249 |
| | msIC↑ | 0.036 | 0.039 | 0.029 | 0.011 | 0.020 | 0.046 | -0.022 | 0.026 | 0.005 | 0.007 | 0.012 | 0.009 | 0.002 | 0.019 | 0.018 | 0.004 |
| | msIR↑ | 0.101 | 0.127 | 0.116 | 0.057 | 0.090 | 0.129 | -0.187 | 0.079 | 0.006 | 0.025 | 0.045 | 0.029 | -0.003 | 0.066 | 0.062 | 0.014 |
| BTCF | MSE↓ | 0.183 | 0.183 | 0.186 | 0.187 | 0.183 | 0.184 | 0.184 | 0.208 | 0.184 | 0.187 | 0.215 | 0.228 | 0.209 | 0.221 | 0.245 | 0.429 |
| | MAE↓ | 0.219 | 0.217 | 0.220 | 0.221 | 0.218 | 0.221 | 0.224 | 0.231 | 0.229 | 0.218 | 0.271 | 0.271 | 0.251 | 0.294 | 0.304 | 0.313 |
| | msIC↑ | 0.162 | 0.168 | 0.159 | 0.151 | 0.159 | 0.158 | 0.151 | 0.145 | 0.151 | 0.162 | 0.159 | 0.127 | 0.114 | 0.150 | 0.089 | 0.000 |
| | msIR↑ | 0.812 | 0.814 | 0.782 | 0.749 | 0.819 | 0.802 | 0.798 | 0.680 | 0.813 | 0.794 | 0.861 | 0.554 | 0.527 | 0.776 | 0.383 | 0.001 |

msIR reflects the ratio between the effective correlation (represented by msIC) achieved by the model and the correlation "noise" arising from the dynamic changes of the time series (as reflected by the standard deviation of the msIC sequence). A higher value indicates that the model achieves high and stable correlation across different samples (large msIC and small standard deviation), suggesting better reliability in the temporal forecasting performance. A lower value may indicate that although the model performs well in some samples (with large msIC at certain points), there is high variability across different samples, which reflects poor model reliability or stability.

## 5 Experiment

To bridge the gap between real-world financial time series data and cutting-edge time series models, we employ over 10 advanced time series models and conduct extensive experimental tests across three financial time series scenarios: Multivariate-to-Multivariate Forecasting, Multivariate-to-Univariate Forecasting, and Multivariate-to-Partial Forecasting. These time series models include: TimeMixer[46], Koopa[30], iTransformer[29], PSformer[47], TiDE[14], PatchTST[38], DLinear[56], Stationary[31], TSMixer[12], TimesNet[52], FEDformer[60], Autoformer[53], Crossformer[57], Transformer[44], Informer[59], and a Naive model.

By designing prediction tasks and evaluation protocols aligned with real-world financial applications, our work provides a comprehensive benchmark of current SOTA time series models in practical financial settings. Additionally, we introduce in-depth insights from AI-driven methodologies to advance financial time series forecasting. Detailed experimental setups and supplementary analyses are documented in Appendix B.1.

### 5.1 Multivariate-to-Multivariate Forecasting

**Setup**. Multivariate forecasting of multivariate tasks is widely used in time series forecasting experiments, such as weather forecasting or electricity forecasting. We extensively evaluated the performance of 16 time series models on these tasks across three datasets, taking into account the non-stationary nature and low signal-to-noise ratio of financial time series, as well as the characteristics of trading days. For each dataset, we selected four different forecasting lengths $H \in \{5, 21, 63, 126\}$, and each task was run 5 times to ensure the robustness of the experiments. We used MSE and MAE as metrics to measure the error between predicted and actual values, and msIC and msIR as metrics to measure time series correlation.

**Results**. The Table 1 presents the average performance of these models on each dataset, with the complete experimental results detailed in Table 8. From the model comparison, it is evident that no single model demonstrates absolute superiority across all metrics on every dataset; however, comparative advantages do exist. Among them, PSformer, TimeMixer, TiDE, and PatchTST exhibit competitive performance in the majority of tasks, with PSformer achieving the best performance in 8 out of 12 instances. It is noteworthy that although earlier models, such as Transformer and FEDformer, are not competitive in terms of error metrics, they show competitiveness in correlation metrics on some datasets, providing an additional dimension for model evaluation. The Naive model, which simply repeats the last value of the input time series, almost lacks predictive correlation, yet its error metrics remain at a low level, even surpassing some cutting-edge time series models. This phenomenon is widely observed in non-stationary and non-periodic time series forecasting.

Table 2: GSMI Timing Strategy Statistic Metrics. Each metric marked with ↑ signifies that higher values are preferred, while those marked with ↓ indicate that lower values are preferred. A detailed explanation of each metric is provided in Appendix C.2.

| Metric | PSformer | TimeMixer | Koopa | iTransformer | TiDE | PatchTST | DLinear | Stationary | TSMixer | TimesNet | FEDformer | Autoformer | Crossformer | Transformer | Informer | Naive |
|---|---|---|---|---|---|---|---|---|---|---|---|---|---|---|---|---|
| Annual return↑ | 17.87% | 9.17% | 8.74% | 9.76% | 7.65% | 6.66% | 15.23% | 11.43% | 9.05% | 11.35% | 8.99% | 8.84% | 3.45% | -0.06% | 8.44% | -0.52% |
| Cumulative returns↑ | 133.68% | 57.30% | 54.11% | 61.76% | 46.30% | 39.52% | 107.94% | 74.80% | 56.37% | 74.18% | 55.97% | 54.89% | 19.14% | -0.31% | 51.96% | -2.67% |
| Annual volatility↓ | 14.61% | 12.07% | 15.79% | 15.59% | 17.68% | 15.93% | 14.52% | 17.15% | 16.60% | 16.85% | 17.13% | 13.70% | 12.74% | 17.84% | 14.71% | 12.41% |
| Sharpe ratio↑ | 1.2 | 0.79 | 0.61 | 0.68 | 0.51 | 0.49 | 1.05 | 0.72 | 0.61 | 0.72 | 0.59 | 0.69 | 0.33 | 0.09 | 0.62 | 0.02 |
| Calmar ratio↑ | 0.83 | 0.49 | 0.31 | 0.38 | 0.27 | 0.23 | 0.73 | 0.4 | 0.26 | 0.4 | 0.32 | 0.53 | 0.12 | 0 | 0.41 | -0.02 |
| Stability↑ | 0.98 | 0.77 | 0.94 | 0.93 | 0.81 | 0.78 | 0.93 | 0.93 | 0.88 | 0.95 | 0.6 | 0.77 | 0 | 0.11 | 0.8 | 0.19 |
| Max drawdown↓ | -21.63% | -18.73% | -27.81% | -25.68% | -28.52% | -28.52% | -20.75% | -28.52% | -34.33% | -28.52% | -28.52% | -16.73% | -29.45% | -33.93% | -20.75% | -31.35% |
| Omega ratio↑ | 1.41 | 1.22 | 1.19 | 1.2 | 1.15 | 1.16 | 1.36 | 1.24 | 1.2 | 1.23 | 1.19 | 1.21 | 1.07 | 1.02 | 1.19 | 1 |
| Sortino ratio↑ | 1.77 | 1.14 | 0.89 | 0.98 | 0.71 | 0.67 | 1.62 | 1.03 | 0.85 | 1.04 | 0.83 | 0.97 | 0.44 | 0.12 | 0.9 | 0.03 |

## 5.2 Multivariate-to-Univariate Forecasting

**Setup**. Multivariate forecasting of univariate time series is a crucial experimental setup in time series prediction, with a wide range of practical applications. We not only evaluate the performance of models from the perspective of time series forecasting but also construct various investment strategies based on the application scenarios of different financial datasets, such as timing trading and long-short trading, and assess the performance of these models within the investment strategies.

**Results**. In the performance evaluation presented in Table 9 and Table 10, the Naive model maintains lower MSE and MAE losses in most cases. Additionally, PSformer, PatchTST, and DLinear exhibit relatively smaller losses. The Naive model achieves the lowest error metrics on the GSMI and BTCF datasets, which is related to the higher difficulty in forecasting price sequences. In terms of correlation metrics, PSformer, Stationary, and DLinear demonstrate more competitive performance. Overall, PSformer performs the best or second-best in 9 out of 12 metrics. While the evaluation metrics effectively showcase the models' forecasting performance, Figure 8 visually illustrates the market timing performance (following the strategy pipeline detailed in Figure 3) of the models on GSMI, and Table 2 quantifies the comparison of strategy statistical metrics across different models. For further univariate experiments and relative discussion, please refer to Appendix B.3. These results provide a broader perspective on the comparison of model performance and potential applications in the financial domain.

## 5.3 Multivariate-to-Partial Forecasting

**Setup**. Multivariate prediction of partial variables is not currently the mainstream experimental setup in time series forecasting. However, [49] discusses the importance of this practical scenario. In this work, we set the closing prices of 20 indices in the GSMI dataset as the target variables for prediction. We evaluate the performance of the models on the GSMI dataset. Additionally, we construct a portfolio selection strategy and backtest the return performance while holding different numbers of indices simultaneously.

**Results**. The Table 11 and Table 12 shows the performance of predicting partial variables. In terms of error metrics, the Naive and PatchTST models perform better, while PSformer and Informer show better correlation metrics. From the portfolio selection backtest plot in Figure 10, no model consistently achieves the highest cumulative return across different numbers of indices held. However, in most cases, the cumulative returns of these models are higher than the average return curve of the 20 indices, and this trend becomes more pronounced as the number of indices held decreases.

## 6 Conclusion

This study bridges the gap between advanced time series forecasting models and practical financial applications. We construct specialized financial datasets capturing distinct market dynamics across global indices, derivatives, and cryptocurrency markets, through msIC and msIR metrics to quantify temporal correlations in multi-step forecasting tasks. Besides, extensive strategy evaluations and visualizations validate the effectiveness and potential of advanced models in real-world financial deployment. Future work will explore integrating large foundation models and agent-based systems with broader financial time series analysis tasks.

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

## Appendices Contents

# A Dataset Details

Table 3: Summary of new datasets.

| Dataset | GSMI | OPTION | BTCF |
|---|---|---|---|
| Range | (2000-01, 2024-12) | (2024-04, 2024-12) | (2020-01, 2024-11) |
| Variate | 100 | 22 | 12 |
| Samples | 6533 | 37431 | 43014 |
| Frequency | Daily | Minutely | Hourly |
| Finance Aera | Stock Indices | Option | Future & Spot |
| Predict Length | {5,21,63,126} | {5,21,63,126} | {5,21,63,126} |
| Dataset Size | (4573, 654 ,1306) | (26201, 3744, 7486) | (30109, 4303, 8602) |

## A.1 GSMI Dataset

### A.1.1 Discription

The GSMI dataset includes 20 major indices from global stock markets, recording daily price and trading volume data for these indices over nearly 20 years, from 2005 to 2024. We have introduced each index in the table below. Index prediction has always been an important and challenging issue, whether it is forecasting the future price trends of the indices or predicting market trading volumes, both of which have significant implications for investment and trading behavior. Research has shown that there are time-varying and complex relationships between global market indices [55, 3, 9, 54]. These complex interrelationships between indices highlight the challenges of forecasting using global index datasets. By leveraging state-of-the-art time-series AI models, we aim to explore the frontier of this important issue and provide a more challenging dataset for time-series prediction, which will better demonstrate the performance differences between different models.

Table 4: Global Stock Market Indices and Descriptions

| Index | Description |
|---|---|
| GSPTSE | S&PTSX Composite Index. Represents the Canadian stock market, reflecting the performance of major companies listed in Canada. |
| ATX | Austrian Traded Index. Reflects the performance of the 30 largest companies listed on the Austrian stock exchange. |
| MXX | Mexbol Index. Represents the main stock index of Mexico, composed of key companies listed on the Mexican Stock Exchange. |
| AEX | Amsterdam Exchange Index. Reflects the performance of major companies listed on the Amsterdam Stock Exchange in the Netherlands. |
| SSMI | Swiss Market Index. Represents the Swiss stock market, composed of the 20 largest companies listed on the Swiss stock exchange. |
| AORD | Australian Ordinaries Index. Tracking the performance of all ordinary shares listed on the Australian Stock Exchange (ASX). It is one of the key indices representing the Australian stock market. |
| SPX | S&P 500 Index. A major American stock index, representing the performance of 500 large-cap companies listed in the U.S. |
| IXIC | NASDAQ Composite Index. Represents the performance of all stocks listed on the NASDAQ stock exchange, covering thousands of companies. |
| DJI | Dow Jones Industrial Average. A key American stock index, composed of 30 major industrial companies, serving as a core market indicator in the U.S. |
| FTSE | FTSE 100 Index. Represents the 100 largest companies listed on the London Stock Exchange. |

| Index | Description |
|---|---|
| GDAXI | DAX Index. Represents the performance of the 30 largest companies listed on the Frankfurt Stock Exchange in Germany. |
| FCHI | CAC 40 Index. Represents the performance of 40 major companies listed on the Paris Stock Exchange in France. |
| JKSE | Jakarta Stock Exchange Composite Index. Represents the performance of companies listed on the Indonesia Stock Exchange, reflecting the overall market of Indonesia. |
| KS11 | KOSPI Index. Represents the performance of the Korean stock market, primarily consisting of companies listed on the Korea Stock Exchange. |
| N225 | Nikkei 225 Index. A well-known Japanese stock index composed of 225 major companies listed on the Tokyo Stock Exchange. |
| KLSE | Kuala Lumpur Stock Exchange Composite Index. Represents the performance of companies listed on the Malaysian stock exchange, reflecting the overall market in Malaysia. |
| SENSEX | Bombay Stock Exchange Sensitive Index. Represents the performance of 30 major companies listed on the Bombay Stock Exchange in India. |
| HSI | Hang Seng Index. Represents the performance of the 50 largest companies listed on the Hong Kong Stock Exchange. |
| SSE | Shanghai Stock Exchange Composite Index. Represents the performance of all stocks listed on the Shanghai Stock Exchange, which is one of China's primary stock exchanges. It is a broad market index for Chinese stocks. |
| SZSE | Shenzhen Stock Exchange Composite Index. Represents the performance of all stocks listed on the Shenzhen Stock Exchange, another major stock exchange in China. This index has a heavier emphasis on smaller and more tech-oriented companies compared to the SSE. |

### A.1.2 Dataset Visualization

The Figure 4 illustrates the preprocessed time series of 20 GSMI indices alongside their raw counterparts. The original series Figure 4a exhibit significant variations in magnitude across indices (spanning four orders of magnitude), complicating cross-variable comparisons. Additionally, the cumulative compounding effect from daily returns amplifies volatility and creates long-range temporal dependencies.

In contrast, the processed series Figure 4b demonstrate three critical improvements: 1) Magnitude Alignment: Normalization ensures all indices operate within comparable scales; 2) Stationarity Enhancement: Logarithmic transformation mitigates non-stationary trends; 3) Improved Properties of Log-Price Series: The processed series exhibit smoother trends and reduced volatility, making them more suitable for analysis and modeling.

### A.1.3 Cross-Variables Comparison

The Figure 5 shows the comparison of the open, high, low, and close price series of the indices before and after processing. It can be observed that after processing, the relative relationships among different prices are preserved, and the magnitudes of the time series of different indices are comparable. This helps the model capture spatio-temporal information.

### A.2 OPTION Dataset

### A.2.1 Discription

Options are an important financial derivative that gives investors the right, but not the obligation, to buy or sell an underlying asset at a predetermined price within a specified time frame. The pricing of options, as well as the prediction of implied volatility and Greek letters, is of significant importance and presents a considerable challenge. Based on this, we have compiled a dataset of the CSI 300 ETF options in the Chinese financial market. Given that options have relatively short expiration periods, this presents challenges for time-series forecasting. Therefore, we have collected minute-frequency

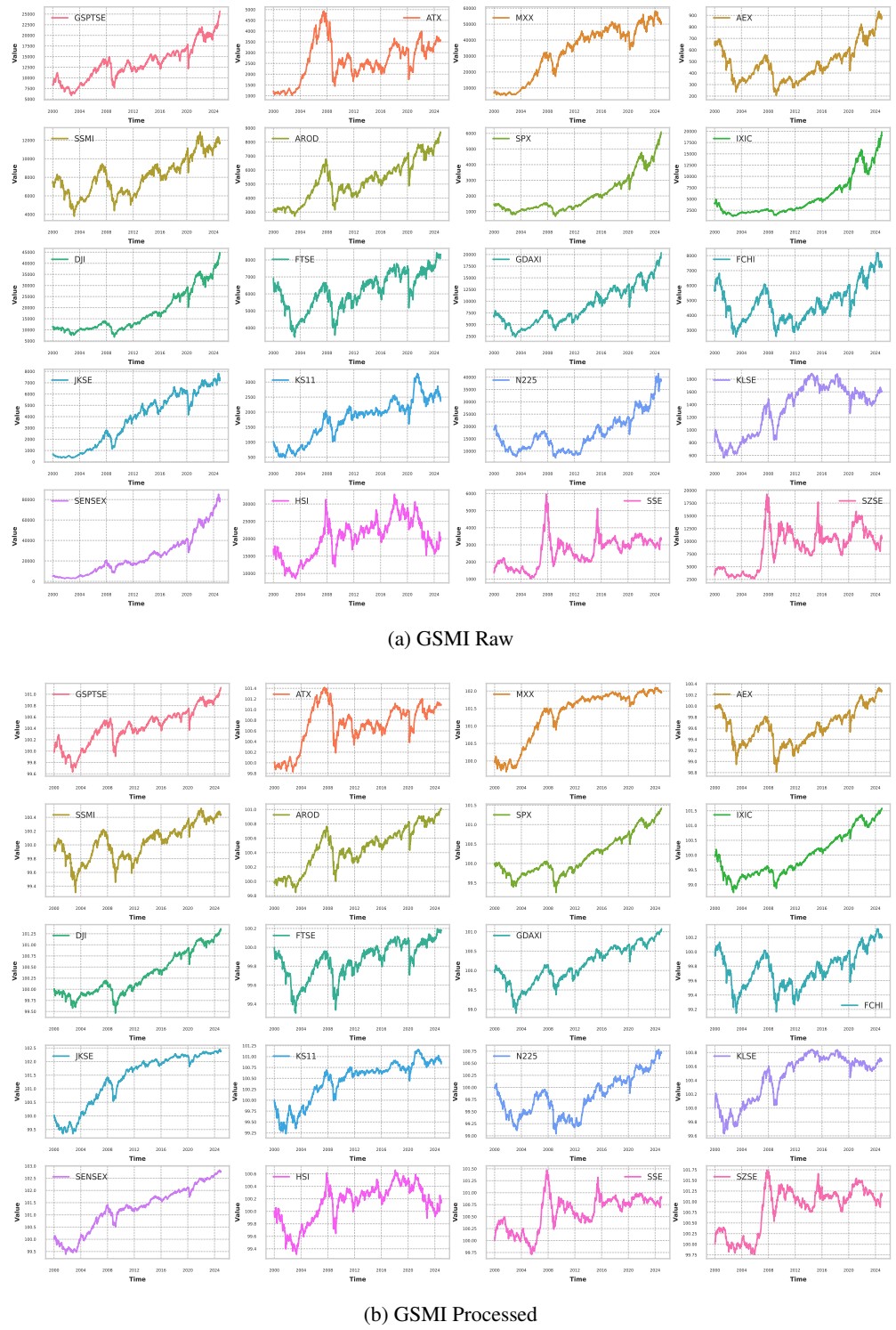

(a) GSMI Raw

(b) GSMI Processed

Figure 4: GSMI Dataset Preprocessing: Raw vs. Processed.

data for this option, which aids in the study of intraday fluctuations in option prices and implied volatility. Specifically, we have collected relevant metrics for both call and put options of the CSI 300 ETF, which together form the variables of this dataset. The data spans from April 25, 2024, to

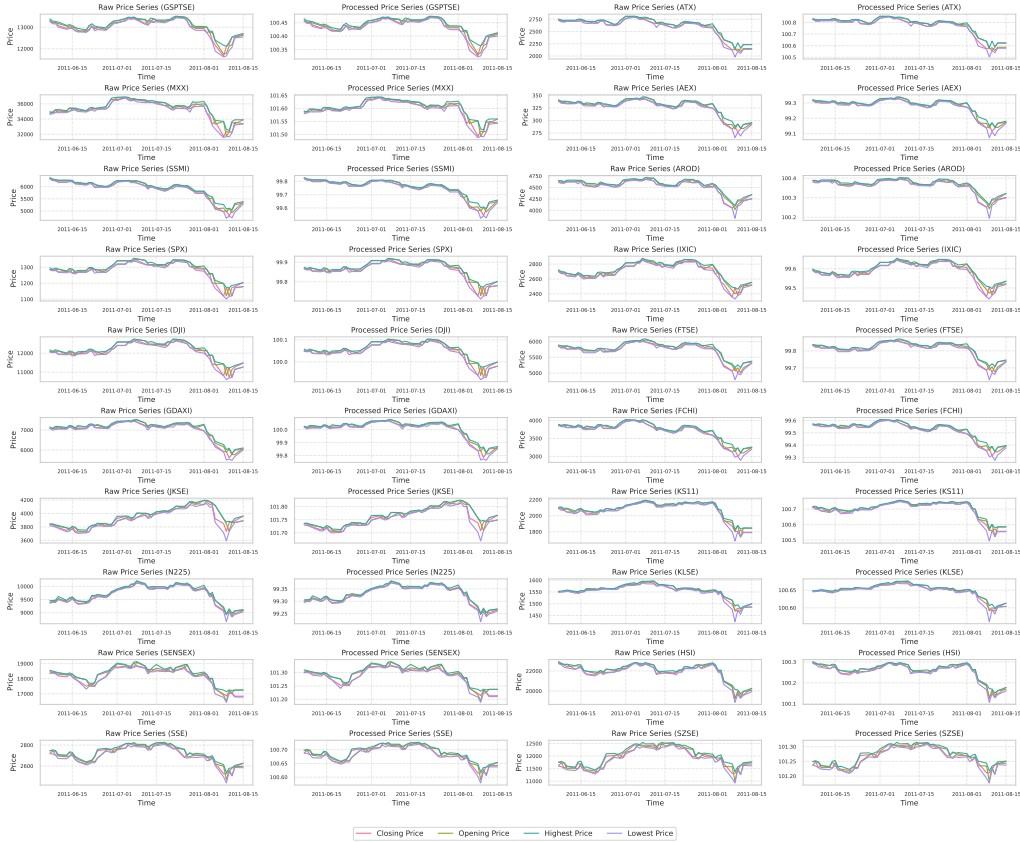

Figure 5: GSMI OHLC Series comparison. This image illustrates the relative relationships between the open, close, high, and low prices of the GSMI indices before and after transformation. The relative relationships remain consistent between the two.

December 13, 2024, with the underlying asset being the CSI 300 ETF, and both options having a strike price of 3.7. We have also derived volatility and Greek letter metrics from the basic data.

### A.2.2 Implied Volatility Calculation

Implied Volatility (IV) refers to the volatility level implied by the option market price, which is typically derived by solving the Black-Scholes model in reverse. The specific calculation method involves solving for the implied volatility using the Newton-Raphson iterative algorithm in the option pricing formula.

Assume the Black-Scholes option pricing formula is:

$$C = S_0 N(d_1) - X e^{-rT} N(d_2),$$

where:

- $C$ is the market price of the option.
- $S_0$ is the current spot price of the underlying asset.
- $X$ is the strike price of the option.
- $r$ is the risk-free interest rate.
- $T$ is the time to expiration of the option.
- $N(d_1)$ and $N(d_2)$ are the cumulative distribution functions of the standard normal distribution.
- $d_1 = \frac{\ln(S_0/X) + (r + \frac{\sigma^2}{2})T}{\sigma\sqrt{T}}$

- $d_2 = d_1 - \sigma\sqrt{T}$

The process of calculating the implied volatility involves substituting the actual price of the market option into the Black-Scholes formula and then solving for the volatility $\sigma$ using the Newton-Raphson iteration method.

### A.2.3 Greek Letters Calculation

Greek letters are used to measure the sensitivity of option prices to changes in different factors. Commonly used Greek letters include Delta, Theta, Gamma, Vega, and Rho. Their calculation formulas are as follows:

- Delta ($\Delta$): The sensitivity of the option price to changes in the price of the underlying asset, calculated as:
$$\Delta = \frac{\partial C}{\partial S_0} = N(d_1),$$
where $N(d_1)$ is the cumulative distribution function of the standard normal distribution, representing the sensitivity of the option price to changes in the price of the underlying asset.

- Theta ($\Theta$): The sensitivity of the option price to the passage of time, calculated as:
$$\Theta = -\frac{S_0 N'(d_1)\sigma}{2\sqrt{T}} - rXe^{-rT}N(d_2),$$
where $N'(d_1)$ is the probability density function of the standard normal distribution, representing the sensitivity of the option price to the passage of time.

- Gamma ($\Gamma$): The second-order sensitivity of the option price to changes in the price of the underlying asset, calculated as:
$$\Gamma = \frac{\partial^2 C}{\partial S_0^2} = \frac{N'(d_1)}{S_0\sigma\sqrt{T}},$$
where $N'(d_1)$ is the probability density function of the standard normal distribution, representing the second-order reaction of the option price to changes in the price of the underlying asset.

- Vega ($\nu$): The sensitivity of the option price to changes in volatility is calculated as
$$\nu = S_0\sqrt{T}N'(d_1),$$
where $N'(d_1)$ is the probability density function of the standard normal distribution, representing the sensitivity of the option price to changes in implied volatility.

- Rho ($\rho$): The sensitivity of the option price to changes in the risk-free interest rate, calculated as:
$$\rho = XTe^{-rT}N(d_2).$$
It represents the sensitivity of the option price to changes in interest rates.

### A.2.4 Historical Volatility Calculation

Historical Volatility (HV) is the standard deviation of asset price changes over a specified time period, used to measure the historical volatility of the underlying asset. The calculation formula is the following.

$$HV = \sqrt{\frac{1}{N-1}\sum_{i=1}^{N}(r_i - \bar{r})^2},$$

where $r_i$ is the return on day $i$, typically the daily logarithmic return: $r_i = \ln\left(\frac{P_i}{P_{i-1}}\right)$, where $P_i$ is the closing price on day $i$, and $P_{i-1}$ is the closing price of the previous day. $\bar{r}$ is the average return over the $N$ days. $N$ is the number of days in the calculation period.

By calculating the historical volatility, we can measure the intensity of asset price fluctuations over the past period, providing a basis for option pricing and risk management.

Table 5: Variables in the OPTION Dataset and Their Descriptions

| Variable | Description |
|---|---|
| **Call Options** | |
| close_call | Closing price of the call option |
| volume_call | Trading volume of the call option |
| iv_call | Implied volatility of the call option |
| his_call | Historical volatility of the call option |
| delta_call | Delta (sensitivity to underlying price) of the call option |
| theta_call | Theta (sensitivity to time decay) of the call option |
| gamma_call | Gamma (sensitivity of delta to price) of the call option |
| vega_call | Vega (sensitivity to volatility) of the call option |
| rho_call | Rho (sensitivity to interest rates) of the call option |
| **Put Options** | |
| close_put | Closing price of the put option |
| volume_put | Trading volume of the put option |
| iv_put | Implied volatility of the put option |
| his_put | Historical volatility of the put option |
| delta_put | Delta (sensitivity to underlying price) of the put option |
| theta_put | Theta (sensitivity to time decay) of the put option |
| gamma_put | Gamma (sensitivity of delta to price) of the put option |
| vega_put | Vega (sensitivity to volatility) of the put option |
| rho_put | Rho (sensitivity to interest rates) of the put option |
| **Common Metrics** | |
| risk_free | Risk-free rate (proxied by 1-Year Shibor) |
| close_base | Closing price of the underlying asset |
| volume_base | Trading volume of the underlying asset |
| t | Time to expiration of the option (in years) |

### A.2.5 OPTION Variables Time Series

The Figure 6 shows the time series plots of each variable in the dataset, derived from real option data. The series exhibit significant fluctuations rooted in the sharp volatility of underlying asset prices and their influence on temporal pattern variations, which holds critical research significance for practical financial pricing problems.

## A.3 BTCF Dataset

### A.3.1 Discription

The BTCF dataset is sourced from the Binance platform and contains hourly frequency data for Bitcoin spot and futures contracts, spanning from January 1, 2020, to November 30, 2024. A notable feature of this dataset is that it records both price and volume variables for the Bitcoin spot market and futures market as in Table 6. There is often a close and complex relationship between futures and spot assets, particularly a time-varying priority-lag relationship (i.e., the dynamic relationship between futures and spot prices). This type of relationship is a widely studied topic in financial markets, especially in the digital currency market, where the futures market often leads the spot market to some extent, and the interaction between price fluctuations and trading volumes also exhibits time-varying characteristics.

This dataset provides rich data support for studying the time-varying relationship between spot and futures markets, and is particularly useful for exploring the performance of state-of-the-art time series models in the digital currency domain. By analyzing this dataset, researchers can delve into the interaction between Bitcoin spot and futures contracts, further investigate their priority-lag relationship, and model and predict the behavior of digital currency markets.

The specific variables in the dataset are listed as follows:

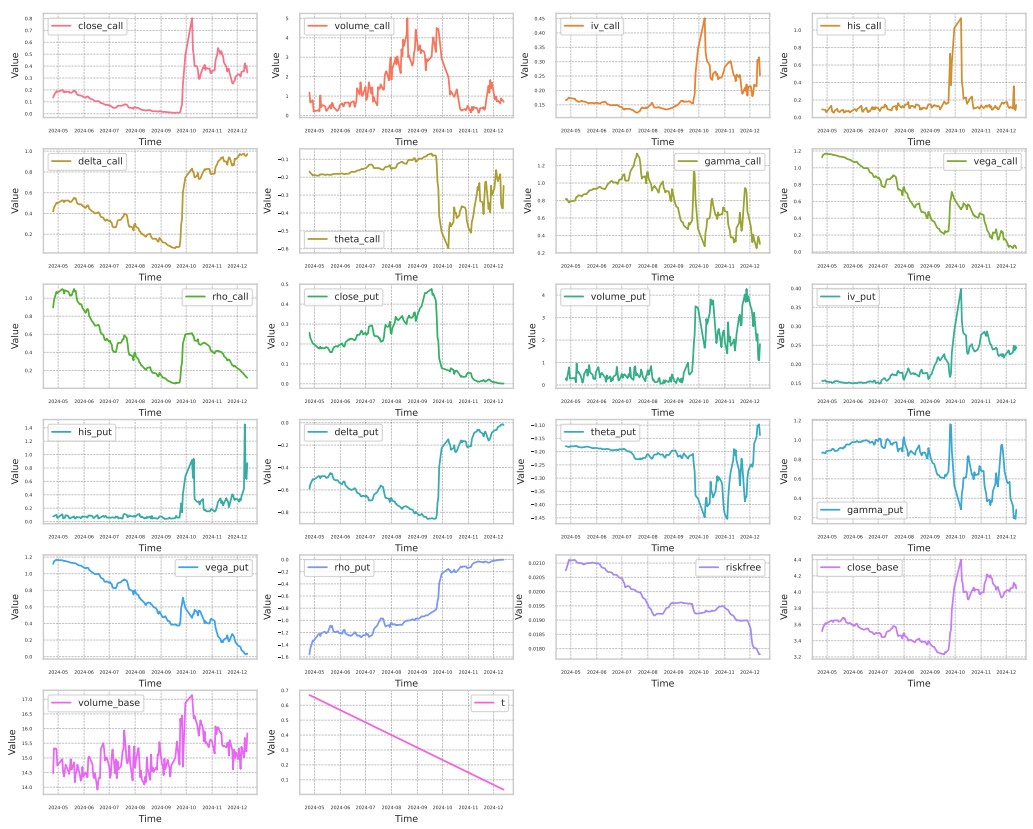

Figure 6: Visualization for full OPTION Time Series Variables

Table 6: BTCF Variables both in Futures and Spot Market Fields

| Metric | Futures Market | Spot Market |
|---|---|---|
| Opening Price | `open_future` | `open_spot` |
| High Price | `high_future` | `high_spot` |
| Low Price | `low_future` | `low_spot` |
| Closing Price | `close_future` | `close_spot` |
| Trading Volume | `volume_future` | `volume_spot` |
| Taker Buy Volume | `taker_buy_volume_future` | `taker_buy_volume_spot` |

This dataset is constructed to provide empirical evidence for the study of priority-lag effects, time-varying relationships, and other phenomena in the digital currency field, and to support further exploration of time-series models' performance on such financial data.

### A.3.2 Preprocessing Comparison

The Figure 7 compares the raw and preprocessed OHLCV (Open-High-Low-Close-Volume) time series of Bitcoin spot prices. For cryptocurrency time series analysis, preprocessing is crucial to address inherent non-stationarity and extreme magnitude variations. Common techniques include normalization to align price scales, logarithmic transformations to stabilize volatility clustering, and detrending operations to mitigate non-stationary price dynamics – all essential for improving model robustness in downstream tasks such as risk assessment or price forecasting.

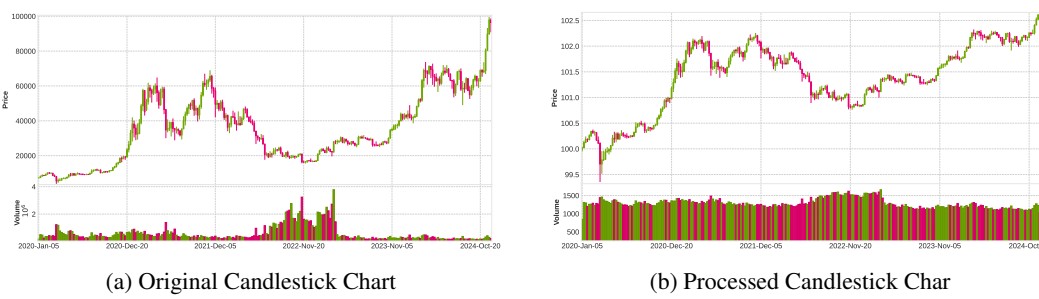

| (a) Original Candlestick Chart | (b) Processed Candlestick Char |

Figure 7: Bitcoin Candlestick Charts: Original and Processed.

## A.4 Statistic Comparison with Mainstream Time Series Datasets

The statistical summary of those three financial domain datasets we constructed as well as the nine mainstream datasets, is reported in Table 7. When the $p\_value$ exceeds 5%, it can be considered that the time series of the dataset is non-stationary. Moreover, the larger the $p\_value$ and ADF values, the more non-stationary the time series is. From the table, it can be seen that 7 out of the 9 mainstream datasets are stationary, with the exceptions being ILI and Exchange. ILI has a weekly frequency and only 966 sample points, while Exchange, which comes from the financial domain, exhibits non-stationarity. However, it can only be considered as one aspect of the financial domain. The three additional datasets we provide complement the existing datasets very well and include more comprehensive variables for exploring time-varying relationships across variables.

Table 7: Statistical comparison between the widely used datasets and the our financial datasets.

| Dataset | Range | Variate | Samples | Frequency | Area | Predict Length | Dataset Size | ADF | p_value |
|---------|-------|---------|---------|-----------|------|----------------|--------------|-----|---------|
| ETTh1 | (2016-07, 2018-06) | 7 | 17420 | Hourly | Electricity | {96, 192, 336, 720} | (8545, 2881, 2881) | -5.9089 | 0.0012 |
| ETTh2 | (2016-07, 2018-06) | 7 | 17420 | Hourly | Electricity | {96, 192, 336, 720} | (8545, 2881, 2881) | -4.1359 | 0.0217 |
| ETTm1 | (2016-07, 2018-06) | 7 | 69680 | 15 Minutes | Electricity | {96, 192, 336, 720} | (34465, 11521, 11521) | -14.9845 | 0 |
| ETTm2 | (2016-07, 2018-06) | 7 | 69680 | 15 Minutes | Electricity | {96, 192, 336, 720} | (34465, 11521, 11521) | -5.6636 | 0.003 |
| Electricity | (2016-07, 2019-07) | 321 | 26304 | Hourly | Electricity | {96, 192, 336, 720} | (18317, 2633, 5261) | -8.4448 | 0.0051 |
| Exchange | (1990-01, 2010-10) | 8 | 7588 | Daily | Exchange rate | {96, 192, 336, 720} | (5120, 665, 1422) | -1.9024 | 0.3598 |
| Weather | (2020-01, 2021-01) | 21 | 52696 | 10 Minutes | Weather | {96, 192, 336, 720} | (36792, 5271, 10540) | -26.6814 | 0 |
| Traffic | (2016-07, 2018-07) | 862 | 17544 | Hourly | Transportation | {96, 192, 336, 720} | (12185, 1757, 3509) | -15.0209 | 0 |
| ILI | (2002-01, 2020-06) | 7 | 966 | Weekly | Illness | {24, 36, 48, 60} | (617, 74, 170) | -5.3342 | 0.1691 |
| GSMI (Ours) | (2000-01, 2024-12) | 100 | 6533 | Daily | Stock Indices | {5,21,63,126} | (4573, 654 ,1306) | -1.4458 | 0.5708 |
| OPTION (Ours) | (2024-04, 2024-12) | 22 | 37431 | Minutely | Option | {5,21,63,126} | (26201, 3744, 7486) | -2.6127 | 0.4634 |
| BTCF (Ours) | (2020-01, 2024-11) | 12 | 43014 | Hourly | Future & Spot | {5,21,63,126} | (30109, 4303, 8602) | -4.8034 | 0.4023 |

# B Experimental Details

## B.1 Configurations

**Multivariate Forecasting.** We conducted comprehensive testing on time series forecasting models from the AI top-tier conferences, specifically for multivariate long-term forecasting. The evaluation metrics used in our experiments include MSE, MAE, msIC, and msIR. To ensure the robustness of the experimental results, each task was run five times and the average of these metrics was reported along with the fluctuation of those results.

Due to the large number of models and tasks involved, we were unable to extensively tune hyperparameters to find the best configuration for every model. However, hyperparameter tuning was not the primary focus of this work. In these experiments, PSformer is from **(author?)**, and the other models are from TSLib [48]. Regarding hyperparameters, if a model has corresponding Exchange script configurations, we will adopt those settings as the Exchange dataset is more aligned with the characteristics of our financial datasets. Otherwise, we will use the script parameters from ETT. All the experiments are conducted on NVIDIA A100 80GB GPU.

We set the input length $L = 512$ and tested the output length on four different lengths: $H \in \{5, 21, 63, 126\}$. We chose these specific lengths instead of the more common lengths

$H \in \{96, 192, 336, 720\}$ from mainstream datasets, as financial time series have shorter temporal relevance. Asset price sequences are driven by a game between long and short positions, and long-term predictions imply sustained arbitrage opportunities. With the increasing number of participants, such opportunities diminish, making the application of longer time windows less profitable.

**Univariate and Partial Variable Forecasting.** To better evaluate the performance of these models on the three datasets, we conducted experiments in real-world financial application scenarios. For the GSMI dataset, given the close interconnections between global stock markets, the flow of capital, and the relative value of market pricing, we constructed a strategy for selecting a combination of 20 indices across markets, as well as a market timing strategy for the SPX index. For the BTCF dataset, we designed a testing strategy for Bitcoin futures, considering both long and short timing strategies. For the OPTION dataset, we developed a strategy for predicting and evaluating implied volatility.

For each task, we applied mainstream evaluation systems in the financial domain to provide a professional assessment, bridging the gap between AI time series forecasting models and real-world financial prediction tasks. We uniformly set the input length to 512 and the output length to 5.

## B.2 Additional Multivariate-to-Multivariate Forecasting Results

### B.2.1 Full Results

The Table 8 shows the full results of the multivariate forecasting tasks. Each task was run 5 times, and the average results along with numerical fluctuations are reported.

Table 8: Full Results for multivariate forecasting task. We compare extensive competitive models under four prediction lengths $H \in \{5, 21, 63, 126\}$. The input sequence length $L = 512$, We display the average results with standard deviation obtained on 5 runs with different seed. MAE↓ and MSE↓ are used to evaluate prediction errors, while msIC↑ and msIR↑ are used to assess the temporal correlation of predictions.

| Method | Metric | GSMI 5 | 21 | 63 | 126 | OPTION 5 | 21 | 63 | 126 | BTCF 5 | 21 | 63 | 126 |
|---|---|---|---|---|---|---|---|---|---|---|---|---|---|
| PatchTST | MSE | 0.0613±0.0003 | 0.0851±0.0003 | 0.1380±0.0020 | 0.2123±0.0096 | 0.1779±0.0003 | 0.2174±0.0005 | 0.2753±0.0051 | 0.3520±0.0090 | 0.1500±0.0004 | 0.1752±0.0021 | 0.2019±0.0039 | 0.2077±0.0025 |
| | MAE | 0.1183±0.0015 | 0.1579±0.0010 | 0.2247±0.0014 | 0.2997±0.0058 | 0.1762±0.0004 | 0.2080±0.0005 | 0.2533±0.0020 | 0.2976±0.0023 | 0.1867±0.0008 | 0.2111±0.0047 | 0.2370±0.0042 | 0.2502±0.0036 |
| | msIC | 0.0338±0.0074 | 0.0288±0.0038 | -0.0126±0.0035 | -0.0019±0.0197 | 0.0298±0.0176 | 0.0433±0.0035 | 0.0518±0.0066 | 0.0586±0.0093 | 0.1132±0.0076 | 0.1581±0.0061 | 0.1741±0.0106 | 0.1876±0.0107 |
| | msIR | 0.0639±0.0122 | 0.1112±0.0076 | 0.0939±0.0098 | 0.1542±0.0401 | 0.0593±0.0395 | 0.1164±0.0079 | 0.1582±0.0180 | 0.1830±0.0235 | 0.2136±0.0134 | 0.5551±0.0193 | 1.0419±0.0223 | 1.3979±0.0328 |
| iTransformer | MSE | 0.0713±0.0007 | 0.0961±0.0001 | 0.1448±0.0017 | 0.1930±0.0020 | 0.1832±0.0007 | 0.2257±0.0020 | 0.2918±0.0071 | 0.3643±0.0141 | 0.1542±0.0009 | 0.1774±0.0016 | 0.2004±0.0043 | 0.2141±0.0009 |
| | MAE | 0.1327±0.0012 | 0.1715±0.0006 | 0.2344±0.0022 | 0.2974±0.0021 | 0.1821±0.0003 | 0.2168±0.0025 | 0.2611±0.0033 | 0.3019±0.0043 | 0.1897±0.0012 | 0.2095±0.0011 | 0.2324±0.0011 | 0.2505±0.0006 |
| | msIC | 0.0154±0.0068 | 0.0204±0.0029 | 0.0208±0.0108 | 0.0274±0.0139 | -0.0038±0.0116 | -0.0127±0.0121 | 0.0170±0.0071 | 0.0423±0.0065 | 0.1015±0.0041 | 0.1600±0.0039 | 0.1774±0.0088 | 0.1647±0.0024 |
| | msIR | 0.0304±0.0126 | 0.0802±0.0063 | 0.1301±0.0301 | 0.1177±0.0496 | -0.0078±0.0358 | -0.0027±0.0289 | 0.0910±0.0178 | 0.1475±0.0182 | 0.1923±0.0069 | 0.5621±0.0099 | 1.0289±0.0339 | 1.2147±0.0078 |
| DLinear | MSE | 0.0782±0.0005 | 0.1013±0.0003 | 0.1467±0.0029 | 0.2159±0.0077 | 0.1815±0.0000 | 0.2202±0.0001 | 0.2783±0.0002 | 0.3448±0.0013 | 0.1540±0.0001 | 0.1785±0.0003 | 0.1964±0.0017 | 0.2055±0.0011 |
| | MAE | 0.1549±0.0015 | 0.1886±0.0008 | 0.2474±0.0054 | 0.3293±0.0089 | 0.1798±0.0009 | 0.2135±0.0015 | 0.2572±0.0012 | 0.2998±0.0054 | 0.1907±0.0013 | 0.2140±0.0026 | 0.2389±0.0086 | 0.2506±0.0033 |
| | msIC | 0.0134±0.0107 | 0.0058±0.0071 | -0.0472±0.0143 | -0.1171±0.0127 | -0.0002±0.0267 | -0.0279±0.0123 | -0.0358±0.0119 | -0.0248±0.0295 | 0.1080±0.0040 | 0.1595±0.0135 | 0.1700±0.0079 | 0.1675±0.0117 |
| | msIR | 0.0293±0.0173 | 0.0675±0.0111 | 0.0417±0.0198 | -0.0650±0.0075 | -0.2482±0.3507 | -0.2291±0.1154 | -0.1734±0.0490 | -0.0981±0.1228 | 0.2032±0.0073 | 0.5544±0.0299 | 1.0722±0.0202 | 1.3641±0.0198 |
| TimesNet | MSE | 0.1640±0.0098 | 0.1974±0.0080 | 0.2493±0.0136 | 0.3084±0.0117 | 0.2912±0.0115 | 0.3528±0.0210 | 0.5115±0.0431 | 0.6275±0.0277 | 0.1558±0.0013 | 0.1795±0.0021 | 0.1974±0.0032 | 0.2147±0.0055 |
| | MAE | 0.2638±0.0097 | 0.3002±0.0092 | 0.3553±0.0115 | 0.4104±0.0084 | 0.2687±0.0050 | 0.3027±0.0097 | 0.3674±0.0161 | 0.4096±0.0068 | 0.1882±0.0017 | 0.2094±0.0014 | 0.2279±0.0019 | 0.2473±0.0050 |
| | msIC | 0.0075±0.0101 | 0.0068±0.0100 | -0.0027±0.0119 | 0.0064±0.0263 | 0.0003±0.0140 | 0.0052±0.0104 | 0.0075±0.0074 | 0.0148±0.0076 | 0.0965±0.0021 | 0.1664±0.0067 | 0.1892±0.0071 | 0.1961±0.0217 |
| | msIR | 0.0145±0.0187 | 0.0193±0.0231 | -0.0074±0.0403 | 0.0148±0.0657 | 0.0042±0.0284 | 0.0175±0.0236 | 0.0320±0.0146 | 0.0449±0.0287 | 0.1813±0.0039 | 0.5642±0.0203 | 1.0810±0.0257 | 1.3508±0.0667 |
| Transformer | MSE | 0.9384±0.0240 | 1.0034±0.0248 | 1.0841±0.0278 | 1.1935±0.0724 | 0.8530±0.0165 | 1.0173±0.083 | 1.6728±0.0449 | 1.4612±0.1285 | 0.1745±0.0084 | 0.2103±0.0034 | 0.2390±0.0075 | 0.2605±0.0126 |
| | MAE | 0.7424±0.0135 | 0.7642±0.0108 | 0.7977±0.0112 | 0.8504±0.0290 | 0.7223±0.0093 | 0.7841±0.0343 | 0.9187±0.0453 | 1.0814±0.0621 | 0.2310±0.0157 | 0.2732±0.0077 | 0.3179±0.0095 | 0.3519±0.0237 |
| | msIC | 0.0306±0.0067 | 0.0526±0.0104 | 0.0847±0.0092 | 0.0811±0.0143 | 0.0286±0.0135 | 0.0113±0.0150 | 0.0224±0.0179 | 0.0132±0.0206 | 0.1000±0.0106 | 0.1642±0.0062 | 0.1682±0.0076 | 0.1656±0.0155 |
| | msIR | 0.0593±0.0113 | 0.1764±0.0243 | 0.2673±0.0210 | 0.2688±0.0205 | 0.0977±0.0933 | 0.0293±0.0390 | 0.0922±0.0769 | 0.0441±0.0582 | 0.1882±0.0213 | 0.5674±0.0275 | 1.0117±0.0689 | 1.3351±0.0866 |
| Koopa | MSE | 0.0672±0.0002 | 0.0915±0.0006 | 0.1407±0.0020 | 0.1954±0.0085 | 0.1971±0.0006 | 0.2312±0.0035 | 0.2888±0.0049 | 0.3515±0.0059 | 0.1648±0.0018 | 0.1788±0.0019 | 0.1953±0.0017 | 0.2062±0.0017 |
| | MAE | 0.1302±0.0010 | 0.1690±0.0012 | 0.2305±0.0013 | 0.2912±0.0063 | 0.1938±0.0010 | 0.2203±0.0013 | 0.2626±0.0022 | 0.2990±0.0021 | 0.2018±0.0025 | 0.2108±0.0018 | 0.2247±0.0006 | 0.2421±0.0014 |
| | msIC | 0.0107±0.0047 | 0.0337±0.0042 | 0.0246±0.0083 | 0.0134±0.0109 | 0.0171±0.0133 | 0.0217±0.0034 | 0.0294±0.0063 | 0.0476±0.0047 | 0.0823±0.0027 | 0.1563±0.0078 | 0.1940±0.0077 | 0.2025±0.0055 |
| | msIR | 0.0201±0.0087 | 0.1029±0.0094 | 0.1231±0.0254 | 0.0770±0.0240 | 0.0473±0.0431 | 0.1078±0.0256 | 0.1136±0.0297 | 0.1959±0.0238 | 0.1547±0.0045 | 0.5326±0.0230 | 1.0742±0.0279 | 1.3648±0.0294 |
| TimeMixer | MSE | 0.0739±0.0096 | 0.1083±0.0145 | 0.1352±0.0056 | 0.1793±0.0138 | 0.1822±0.0013 | 0.2254±0.0089 | 0.2751±0.0047 | 0.3392±0.0054 | 0.1537±0.0019 | 0.1771±0.0016 | 0.1940±0.0017 | 0.2056±0.0024 |
| | MAE | 0.1380±0.0151 | 0.1840±0.0178 | 0.2188±0.0060 | 0.2785±0.0152 | 0.1801±0.0018 | 0.2157±0.0069 | 0.2500±0.0021 | 0.2861±0.0024 | 0.1889±0.0046 | 0.2079±0.0015 | 0.2268±0.0011 | 0.2429±0.0007 |
| | msIC | 0.0080±0.0013 | 0.0111±0.0151 | 0.0245±0.0138 | 0.0382±0.0217 | 0.0210±0.0162 | 0.0257±0.0194 | 0.0312±0.0190 | 0.0763±0.0161 | 0.1086±0.0052 | 0.1658±0.0028 | 0.1978±0.0114 | 0.1983±0.0074 |
| | msIR | 0.0159±0.0027 | 0.0393±0.0503 | 0.1130±0.0574 | 0.2120±0.0991 | 0.0498±0.0393 | 0.0923±0.0552 | 0.1186±0.0570 | 0.2488±0.0400 | 0.2053±0.0092 | 0.5656±0.0122 | 1.1129±0.0404 | 1.3736±0.0498 |
| Informer | MSE | 1.0907±0.0137 | 1.1127±0.0237 | 1.1624±0.0159 | 1.2511±0.0220 | 1.4348±0.1210 | 1.6728±0.0508 | 2.3117±0.2339 | 2.9033±0.2956 | 0.1762±0.0035 | 0.2118±0.0078 | 0.2717±0.0197 | 0.3188±0.0238 |
| | MAE | 0.7840±0.0096 | 0.7820±0.0077 | 0.8086±0.0055 | 0.8528±0.0114 | 0.9473±0.0398 | 1.0303±0.0344 | 1.1994±0.0841 | 1.3213±0.0582 | 0.2244±0.0065 | 0.2678±0.0120 | 0.3462±0.0278 | 0.3770±0.0258 |
| | msIC | 0.0126±0.0022 | 0.0216±0.0089 | 0.0175±0.0101 | 0.0460±0.0196 | -0.0005±0.0053 | 0.0072±0.0113 | 0.0260±0.0240 | 0.0388±0.0259 | 0.0586±0.0187 | 0.1357±0.0088 | 0.1038±0.0300 | 0.0566±0.0119 |
| | msIR | 0.0241±0.0041 | 0.0556±0.0203 | 0.0492±0.0298 | 0.1119±0.0488 | -0.0018±0.0125 | 0.0249±0.0367 | 0.1107±0.1075 | 0.1142±0.0825 | 0.1081±0.0334 | 0.4707±0.0277 | 0.5630±0.1884 | 0.3893±0.0425 |
| FEDformer | MSE | 0.2301±0.0005 | 0.2308±0.0044 | 0.2516±0.0040 | 0.2671±0.0069 | 0.5983±0.1138 | 0.5394±0.0449 | 0.5845±0.0178 | 0.6555±0.0110 | 0.2007±0.0036 | 0.2121±0.0027 | 0.2219±0.0054 | 0.2241±0.0021 |
| | MAE | 0.3489±0.0043 | 0.3516±0.0039 | 0.3702±0.0057 | 0.3838±0.0060 | 0.4885±0.0638 | 0.4497±0.0325 | 0.4766±0.0150 | 0.5114±0.0158 | 0.2566±0.0024 | 0.2671±0.0025 | 0.2773±0.0060 | 0.2825±0.0038 |
| | msIC | 0.0166±0.0176 | 0.0264±0.0160 | 0.0513±0.0117 | 0.1141±0.0123 | -0.0020±0.0093 | 0.0017±0.0093 | 0.0151±0.0049 | 0.0342±0.0063 | 0.1082±0.0053 | 0.1725±0.0066 | 0.1768±0.0050 | 0.1782±0.0157 |
| | msIR | 0.0348±0.0298 | 0.0962±0.0346 | 0.2210±0.0454 | 0.5079±0.0258 | -0.0019±0.0186 | 0.0082±0.0240 | 0.0571±0.0140 | 0.1155±0.0257 | 0.2073±0.0101 | 0.6044±0.0179 | 1.1347±0.0254 | 1.4973±0.0570 |
| Crossformer | MSE | 0.5257±0.0344 | 0.6349±0.0186 | 0.7387±0.0399 | 0.9539±0.0169 | 0.2559±0.0081 | 0.3484±0.0094 | 0.4341±0.0273 | 0.4341±0.0331 | 0.1544±0.0003 | 0.1830±0.0010 | 0.2118±0.0045 | 0.2885±0.0019 |
| | MAE | 0.4400±0.0100 | 0.5045±0.0058 | 0.5752±0.0181 | 0.6822±0.0029 | 0.2299±0.0084 | 0.2805±0.0124 | 0.3695±0.0117 | 0.4087±0.0331 | 0.1978±0.0007 | 0.2254±0.0026 | 0.2586±0.0016 | 0.3209±0.0043 |
| | msIC | -0.0035±0.0128 | -0.0014±0.0042 | 0.0051±0.0044 | 0.0146±0.0091 | 0.0002±0.0244 | 0.0012±0.0197 | 0.0087±0.0066 | -0.0035±0.0210 | 0.0998±0.0052 | 0.1527±0.0043 | 0.1550±0.0068 | 0.0489±0.0093 |
| | msIR | -0.0043±0.0218 | -0.0048±0.0181 | 0.0448±0.0339 | 0.1660±0.0751 | -0.0941±0.2977 | -0.0422±0.2148 | 0.1315±0.0907 | -0.0082±0.1389 | 0.1864±0.0091 | 0.5091±0.0144 | 0.9140±0.0480 | 0.4986±0.0403 |
| Autoformer | MSE | 0.2124±0.0229 | 0.2464±0.0241 | 0.2872±0.0193 | 0.3783±0.0713 | 0.4637±0.0063 | 0.4945±0.0307 | 0.7675±0.2447 | 0.7406±0.2835 | 0.1953±0.0053 | 0.2175±0.0041 | 0.2411±0.0047 | 0.2600±0.0068 |
| | MAE | 0.3328±0.0184 | 0.3574±0.0205 | 0.3908±0.0159 | 0.4493±0.0300 | 0.3994±0.0053 | 0.4221±0.0254 | 0.5586±0.1123 | 0.5439±0.1369 | 0.2372±0.0070 | 0.2540±0.0033 | 0.2876±0.0095 | 0.3064±0.0052 |
| | msIC | 0.0249±0.0107 | 0.0314±0.0223 | 0.0380±0.0222 | 0.0632±0.0239 | 0.0113±0.0074 | 0.0005±0.0049 | 0.0039±0.0043 | 0.0196±0.0076 | 0.0914±0.0058 | 0.1352±0.0056 | 0.1329±0.0149 | 0.1483±0.0140 |
| | msIR | 0.0488±0.0177 | 0.0978±0.0442 | 0.1589±0.0505 | 0.2585±0.0762 | 0.0246±0.0129 | 0.0072±0.0103 | 0.0175±0.0129 | 0.0680±0.0244 | 0.1717±0.0100 | 0.4481±0.0277 | 0.6785±0.0734 | 0.9172±0.0493 |
| TSMixer | MSE | 0.1130±0.0018 | 0.1792±0.0086 | 0.2641±0.0341 | 0.3231±0.0120 | 0.3207±0.0078 | 0.4744±0.0729 | 0.7735±0.1028 | 0.9891±0.2649 | 0.1584±0.0047 | 0.1748±0.0033 | 0.1963±0.0063 | 0.2048±0.0034 |
| | MAE | 0.2203±0.0056 | 0.2863±0.0098 | 0.3780±0.0290 | 0.4305±0.0110 | 0.3688±0.0087 | 0.4599±0.0194 | 0.6072±0.0417 | 0.6891±0.1007 | 0.2081±0.0175 | 0.2123±0.0037 | 0.2363±0.0083 | 0.2604±0.0124 |
| | msIC | 0.0131±0.0089 | 0.0078±0.0175 | -0.0217±0.0320 | -0.0847±0.0191 | 0.0009±0.0200 | 0.0105±0.0173 | 0.0283±0.0162 | -0.0198±0.0267 | 0.0875±0.0165 | 0.1607±0.0075 | 0.1715±0.0159 | 0.1835±0.0163 |
| | msIR | 0.0261±0.0169 | 0.0413±0.0438 | -0.0459±0.0742 | -0.3055±0.0227 | -0.1398±0.3228 | 0.0788±0.1369 | 0.1696±0.0977 | -0.0855±0.1188 | 0.1703±0.0288 | 0.5695±0.0219 | 1.0643±0.0669 | 1.4462±0.0471 |
| Stationary | MSE | 0.1326±0.0065 | 0.1682±0.0065 | 0.2061±0.0050 | 0.2421±0.0095 | 0.2599±0.0089 | 0.3386±0.0179 | 0.4505±0.0138 | 0.5327±0.0134 | 0.1562±0.0018 | 0.1889±0.0040 | 0.2281±0.0091 | 0.2599±0.0170 |
| | MAE | 0.2285±0.0075 | 0.2649±0.0063 | 0.3056±0.0054 | 0.3447±0.0089 | 0.2517±0.0052 | 0.2934±0.0059 | 0.3425±0.0066 | 0.3758±0.0034 | 0.1865±0.0023 | 0.2129±0.0029 | 0.2473±0.0061 | 0.2786±0.0116 |
| | msIC | 0.0260±0.0087 | 0.0389±0.0095 | 0.0470±0.0101 | 0.0556±0.0160 | 0.0141±0.0121 | 0.0164±0.0074 | 0.0287±0.0184 | 0.0452±0.0119 | 0.0864±0.0084 | 0.1499±0.0044 | 0.1694±0.0100 | 0.1748±0.0154 |
| | msIR | 0.0499±0.0153 | 0.1207±0.0253 | 0.1746±0.0331 | 0.2021±0.0544 | 0.0280±0.0201 | 0.0575±0.0243 | 0.0864±0.0509 | 0.1433±0.0354 | 0.1617±0.0161 | 0.4968±0.0152 | 0.9453±0.0552 | 1.1147±0.1131 |
| TiDE | MSE | 0.0865±0.0005 | 0.1090±0.0004 | 0.1487±0.0004 | 0.1802±0.0012 | 0.1816±0.0001 | 0.2201±0.0000 | 0.2770±0.0003 | 0.3419±0.0001 | 0.1544±0.0002 | 0.1785±0.0001 | 0.1952±0.0001 | 0.2046±0.0002 |
| | MAE | 0.1536±0.0008 | 0.1842±0.0005 | 0.2306±0.0004 | 0.2735±0.0008 | 0.1775±0.0005 | 0.2077±0.0001 | 0.2467±0.0001 | 0.2836±0.0001 | 0.1891±0.0004 | 0.2099±0.0001 | 0.2292±0.0000 | 0.2434±0.0004 |
| | msIC | 0.0071±0.0073 | 0.0244±0.0031 | 0.0312±0.0028 | 0.0499±0.0025 | -0.0077±0.0048 | -0.0090±0.0010 | 0.0209±0.0007 | 0.0751±0.0026 | 0.1100±0.0016 | 0.1605±0.0008 | 0.1828±0.0001 | 0.1842±0.0012 |
| | msIR | 0.0143±0.0147 | 0.0858±0.0094 | 0.1569±0.0100 | 0.2967±0.0138 | -0.0137±0.0285 | 0.0301±0.0032 | 0.1143±0.0015 | 0.2293±0.0042 | 0.2062±0.0023 | 0.5552±0.0017 | 1.0882±0.0003 | 1.4058±0.0056 |
| PSformer | MSE | 0.0626±0.0003 | 0.0899±0.0004 | 0.1316±0.0010 | 0.1632±0.0006 | 0.1813±0.0001 | 0.2185±0.0010 | 0.2714±0.0025 | 0.3320±0.0015 | 0.1539±0.0007 | 0.1728±0.0011 | 0.1981±0.0030 | 0.2061±0.0009 |
| | MAE | 0.1184±0.0006 | 0.1655±0.0011 | 0.2170±0.0011 | 0.2619±0.0006 | 0.1770±0.0011 | 0.2069±0.0012 | 0.2458±0.0007 | 0.2848±0.0009 | 0.1902±0.0004 | 0.2100±0.0010 | 0.2314±0.0017 | 0.2456±0.0019 |
| | msIC | 0.0369±0.0048 | 0.0528±0.0088 | 0.0743±0.0107 | 0.1072±0.0144 | 0.0084±0.0125 | 0.0092±0.0135 | 0.0442±0.0151 | 0.0802±0.0066 | 0.1141±0.0055 | 0.1472±0.0024 | 0.1847±0.0063 | 0.2019±0.0197 |
| | msIR | 0.0683±0.0074 | 0.1614±0.0160 | 0.2767±0.0281 | 0.3890±0.0204 | -0.0201±0.0687 | 0.0565±0.0254 | 0.1463±0.0274 | 0.2217±0.0147 | 0.2134±0.0094 | 0.5455±0.0070 | 1.0813±0.0148 | 1.4072±0.0367 |
| Naive | MSE | 0.0884±0.0000 | 0.1163±0.0000 | 0.1642±0.0000 | 0.2096±0.0000 | 0.2684±0.0000 | 0.3350±0.0000 | 0.4401±0.0000 | 0.4746±0.0000 | 0.2767±0.0000 | 0.4041±0.0000 | 0.4958±0.0000 | 0.5375±0.0000 |
| | MAE | 0.1651±0.0000 | 0.1642±0.0000 | 0.2233±0.0000 | 0.2725±0.0000 | 0.1872±0.0000 | 0.2354±0.0000 | 0.2706±0.0000 | 0.3116±0.0000 | 0.2419±0.0000 | 0.3003±0.0000 | 0.3427±0.0000 | 0.3677±0.0000 |
| | msIC | -0.0000±0.0007 | -0.0002±0.0007 | -0.0003±0.0002 | 0.0001±0.0003 | 0.0090±0.0012 | 0.0030±0.0004 | 0.0025±0.0004 | 0.0020±0.0002 | -0.0000±0.0018 | 0.0003±0.0007 | 0.0000±0.0005 | 0.0001±0.0002 |
| | msIR | 0.0000±0.0014 | -0.0010±0.0032 | -0.0022±0.0013 | 0.0007±0.0038 | 0.0219±0.0024 | 0.0119±0.0020 | 0.0114±0.0031 | 0.0090±0.0021 | -0.0000±0.0036 | 0.0013±0.0031 | 0.0003±0.0037 | 0.0008±0.0023 |

Table 9: Univarate Results (Part I)

| Dataset | Metric | PSformer | TimeMixer | Koopa | iTransformer | TiDE | PatchTST | DLinear | Stationary |
|---|---|---|---|---|---|---|---|---|---|
| **GSMI** | MSE | 0.0136±0.0005 | 0.1483±0.2124 | 0.0176±0.0003 | 0.0194±0.0010 | 0.0265±0.0013 | 0.0122±0.0021 | 0.0274±0.0009 | 0.0393±0.0088 |
| | MAE | 0.0801±0.0011 | 0.2341±0.1981 | 0.0981±0.0020 | 0.1024±0.0027 | 0.1155±0.0025 | 0.0808±0.0050 | 0.1213±0.0044 | 0.1595±0.0163 |
| | msIC | 0.0826±0.0119 | 0.0131±0.0133 | 0.0266±0.0259 | 0.0240±0.0095 | 0.0108±0.0153 | -0.0055±0.0322 | -0.0446±0.0962 | -0.0247±0.0248 |
| | msIR | 0.1373±0.0185 | 0.0264±0.0281 | 0.0494±0.0474 | 0.0463±0.0188 | 0.0214±0.0305 | -0.0100±0.0612 | -0.0607±0.1644 | -0.0451±0.0452 |
| **OPTION** | MSE | 0.3629±0.0019 | 0.4764±0.0756 | 0.4565±0.0131 | 0.5199±0.0075 | 0.4819±0.0099 | 0.5357±0.0874 | 0.4000±0.0005 | 0.4755±0.0189 |
| | MAE | 0.4292±0.0020 | 0.5038±0.0416 | 0.4882±0.0079 | 0.5252±0.0044 | 0.5029±0.0050 | 0.5382±0.0426 | 0.4609±0.0009 | 0.5044±0.0114 |
| | msIC | 0.0647±0.0033 | 0.0064±0.0072 | 0.0048±0.0047 | 0.0081±0.0053 | 0.0017±0.0031 | 0.0055±0.0068 | 0.0078±0.0026 | 0.0599±0.0137 |
| | msIR | 0.1166±0.0066 | 0.0125±0.0141 | 0.0093±0.0091 | 0.0157±0.0104 | 0.0033±0.0058 | 0.0104±0.0128 | 0.0139±0.0045 | 0.1105±0.0255 |
| **BTCF** | MSE | 0.0004±0.0000 | 0.0003±0.0000 | 0.0003±0.0000 | 0.0003±0.0000 | 0.0004±0.0000 | 0.0003±0.0000 | 0.0004±0.0000 | 0.0003±0.0000 |
| | MAE | 0.0149±0.0005 | 0.0117±0.0007 | 0.0130±0.0006 | 0.0121±0.0002 | 0.0129±0.0001 | 0.0122±0.0004 | 0.0134±0.0001 | 0.0120±0.0009 |
| | msIC | 0.0156±0.0040 | 0.0030±0.0057 | 0.0120±0.0067 | 0.0124±0.0067 | 0.0080±0.0086 | 0.0122±0.0075 | 0.0280±0.0050 | 0.0083±0.0090 |
| | msIR | 0.0263±0.0069 | 0.0060±0.0115 | 0.0228±0.0130 | 0.0232±0.0125 | 0.0160±0.0172 | 0.0236±0.0141 | 0.0453±0.0092 | 0.0146±0.0161 |

Table 10: Univarate Results (Part II)

| Dataset | Metric | TSMixer | TimesNet | FEDformer | Autoformer | Crossformer | Transformer | Informer | Naive |
|---|---|---|---|---|---|---|---|---|---|
| **GSMI** | MSE | 0.1086±0.0267 | 0.0721±0.0220 | 0.1931±0.0174 | 0.1741±0.0328 | 3.3567±2.2070 | 2.1301±0.2142 | 3.0564±0.3746 | 0.0063±0.0000 |
| | MAE | 0.2713±0.0405 | 0.2089±0.0335 | 0.3643±0.0221 | 0.3578±0.0330 | 1.7291±0.0581 | 1.3648±0.0791 | 1.6430±0.1074 | 0.0532±0.0000 |
| | msIC | 0.0093±0.0464 | 0.0072±0.0276 | -0.0027±0.0221 | -0.0112±0.0327 | 0.0566±0.0260 | 0.0163±0.0077 | 0.0525±0.0212 | -0.0083±0.0124 |
| | msIR | 0.0189±0.0865 | 0.0136±0.0536 | -0.0048±0.0397 | -0.0206±0.0579 | 0.1035±0.0583 | 0.0344±0.0161 | 0.1020±0.0404 | -0.0164±0.0245 |
| **OPTION** | MSE | 0.9399±0.2963 | 0.6288±0.0819 | 0.7561±0.0459 | 0.6420±0.0124 | 3.2330±2.1409 | 0.7137±0.1566 | 1.1191±0.4884 | 0.4183±0.0000 |
| | MAE | 0.7381±0.1314 | 0.5757±0.0298 | 0.6287±0.0163 | 0.5900±0.0066 | 1.2665±0.5412 | 0.6349±0.0749 | 0.8070±0.1958 | 0.4328±0.0000 |
| | msIC | 0.0012±0.0015 | 0.0049±0.0034 | 0.0268±0.0125 | 0.0199±0.0129 | 0.0047±0.0045 | 0.0331±0.0128 | 0.0021±0.0028 | 0.0027±0.0035 |
| | msIR | 0.0025±0.0031 | 0.0094±0.0064 | 0.0487±0.0227 | 0.0364±0.0234 | 0.0087±0.0080 | 0.0622±0.0240 | 0.0039±0.0054 | 0.0054±0.0071 |
| **BTCF** | MSE | 0.0005±0.0000 | 0.0003±0.0000 | 0.0128±0.0010 | 0.0059±0.0010 | 0.0035±0.0004 | 0.0041±0.0004 | 0.0097±0.0030 | 0.0002±0.0000 |
| | MAE | 0.0162±0.0005 | 0.0122±0.0002 | 0.0892±0.0025 | 0.0590±0.0057 | 0.0311±0.0021 | 0.0464±0.0038 | 0.0777±0.0153 | 0.0094±0.0000 |
| | msIC | -0.0040±0.0039 | 0.0104±0.0080 | 0.0083±0.0069 | 0.0083±0.0036 | 0.0001±0.0136 | -0.0017±0.0116 | -0.0021±0.0014 | 0.0001±0.0062 |
| | msIR | -0.0078±0.0077 | 0.0186±0.0144 | 0.0143±0.0118 | 0.0144±0.0062 | 0.0002±0.0271 | -0.0035±0.0219 | -0.0041±0.0027 | 0.0001±0.0125 |

## B.3 Additional Multivariate-to-Univariate Forecasting Results

### B.3.1 Full Results

This subsection shows the complete experimental results for univariate forecasting across three datasets. To better present the results, we have divided the results into Table 9 and Table 10, each showcasing the results of eight models. For each task, we highlight the best performance in red font and the second-best performance in blue font with underlining. From these results, we observe that for these challenging non-stationary time series, the Naive model exhibits very low error loss, but its predictive correlation is close to zero. This underscores the importance of correlation metrics, such as msIC and msIR, in the evaluation and application of forecasting models for non-stationary time series.

Additionally, while some models perform well on correlation metrics, their error loss is still high. This indicates that high correlation alone cannot guarantee potential good performance in financial time series applications. A model is more likely to demonstrate predictive capability when both the correlation metrics and error metrics are promising.

### B.3.2 Visualization of GSMI Timing Trading

The Figure 8 illustrates the construction of timing strategies for the S&P 500 index within the GSMI dataset, and visualizes the return curves of each model. PSformer, DLinear and Stationary have achieved competitive performance.

### B.3.3 Visualization of BTCF Long-Short Trading

The Figure 9 constructs long and short strategies for perpetual contracts in the BTCF dataset, visualizing their respective return curves, where PSformer and Koopa demonstrate more competitive performance. Although the cumulative returns are substantial, it should be noted that transaction costs have not been considered in this analysis, as this factor falls outside the primary scope of the current research.

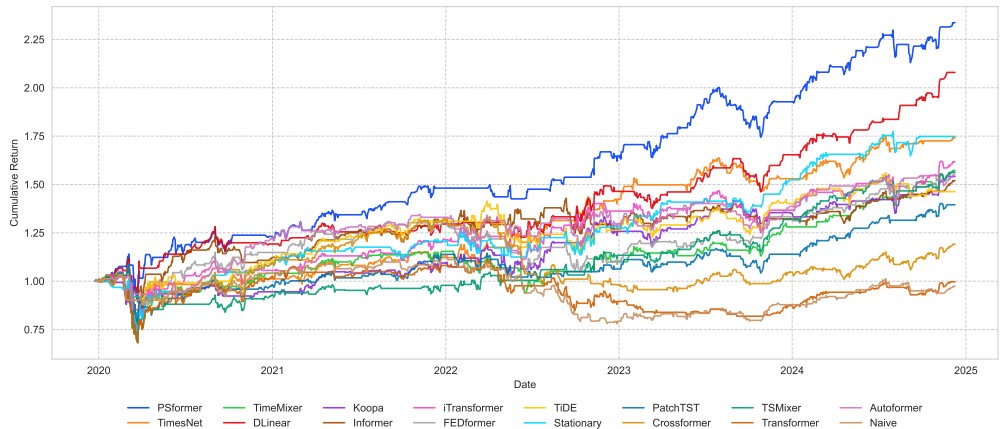

Figure 8: Comparison of Timing returns based on different models. This figure presents performance comparison of 16 models in timing strategies for the SPX index. The vertical axis is normalized (initial net value set to 1), displaying the cumulative return curves generated by each model through timing trading. The position status (holding long positions in the index or maintaining cash positions) is adjusted every five trading days, testing the ability to capturing excess returns through dynamic position management.

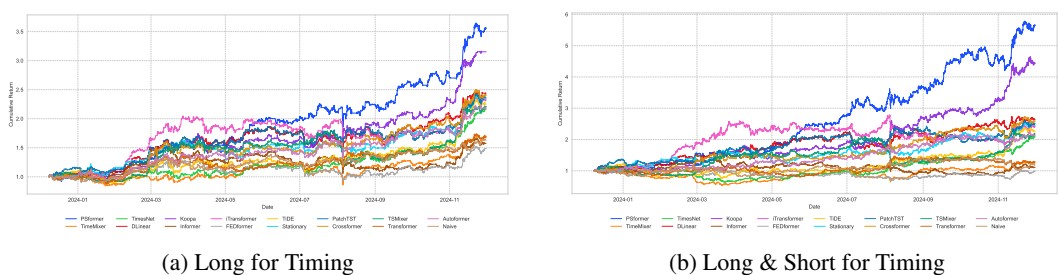

| (a) Long for Timing | (b) Long & Short for Timing |
|:---:|:---:|

Figure 9: BTCF for Timing. These figures present long and short strategies for BTC futures. The left figure illustrates the long-only strategy, where each model generates strategy returns by deciding whether to hold a long position or remain in cash. The right figure demonstrates the long-short strategy, where returns are achieved through dynamically taking either long or short positions.

## B.4 Additional Multivariate-to-Univariate Forecasting Results

### B.4.1 Full Results

Table 11 and Table 12 present the comprehensive experimental results of each model predictive performance on the 20 indices close price from the GSMI dataset.

Table 11: Partial Variates Results (Part I)

| | Metric | PSformer | TimeMixer | Koopa | iTransformer | TiDE | PatchTST | DLinear | Stationary |
|---|---|---|---|---|---|---|---|---|---|
| **GSMI** | MSE | 0.0121±0.0029 | 0.0145±0.0036 | 0.0108±0.0002 | 0.0127±0.0004 | 0.0238±0.0012 | 0.0086±0.0004 | 0.0238±0.0005 | 0.0488±0.0042 |
| | MAE | 0.0701±0.0091 | 0.0784±0.0114 | 0.0682±0.0007 | 0.0724±0.0013 | 0.0998±0.0021 | 0.0602±0.0020 | 0.1026±0.0022 | 0.1609±0.0072 |
| | msIC | 0.0270±0.0039 | 0.0059±0.0115 | 0.0024±0.0015 | 0.0089±0.0105 | -0.0094±0.0069 | -0.0019±0.0185 | -0.0203±0.0288 | 0.0175±0.0105 |
| | msIR | 0.0484±0.0065 | 0.0122±0.0233 | 0.0048±0.0028 | 0.0174±0.0207 | -0.0188±0.0139 | -0.0034±0.0339 | -0.0310±0.0509 | 0.0311±0.0187 |

Table 12: Partial Variates Results (Part II)

| | Metric | TSMixer | TimesNet | FEDformer | Autoformer | Crossformer | Transformer | Informer | Naive |
|---|---|---|---|---|---|---|---|---|---|
| **GSMI** | MSE | 0.0457±0.0118 | 0.0694±0.0070 | 0.1692±0.0221 | 0.1359±0.0141 | 0.5913±0.0478 | 0.6613±0.0386 | 0.9050±0.0538 | 0.0050±0.0000 |
| | MAE | 0.1659±0.0202 | 0.1913±0.0096 | 0.3132±0.0152 | 0.2876±0.0148 | 0.4653±0.0147 | 0.6199±0.0168 | 0.7217±0.0196 | 0.0435±0.0000 |
| | msIC | -0.0043±0.0124 | 0.0024±0.0068 | 0.0006±0.0218 | 0.0081±0.0182 | -0.0077±0.0205 | 0.0029±0.0161 | 0.0203±0.0096 | -0.0004±0.0020 |
| | msIR | -0.0081±0.0247 | 0.0048±0.0119 | 0.0006±0.0375 | 0.0133±0.0301 | -0.0139±0.0351 | 0.0050±0.0303 | 0.0389±0.0181 | -0.0009±0.0039 |

### B.4.2 Visualization of GSMI Portfolio Optimization

The Figure 10 presents the return curves of portfolio optimization strategies applied to the 20 indices in the GSMI dataset. The strategy employs a ranking-based approach, where indices are scored and ranked according to their predicted return potential, with priority given to holding higher-ranked indices. The analysis reveals that the strategy achieves optimal performance when maintaining exposure to a single top-ranked index per rebalancing period. As the portfolio size expands to include more indices, the overall return gradually decreases, converging toward the average return of the 20-index Average benchmark.

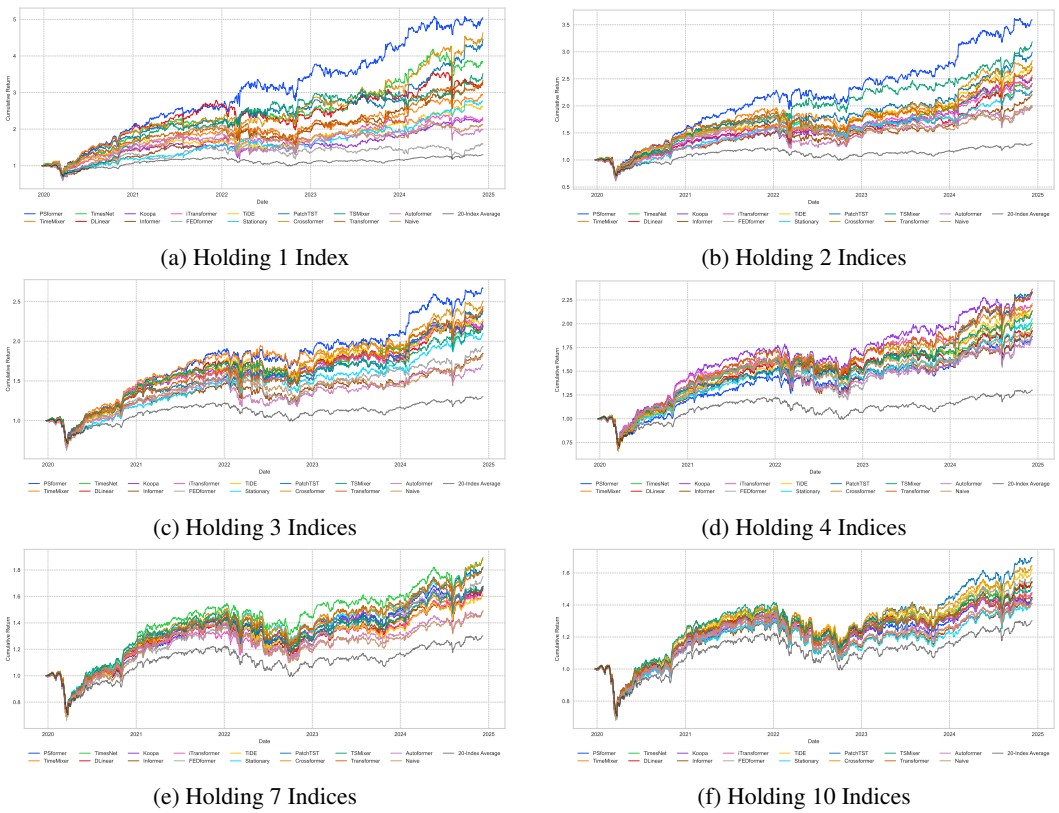

(a) Holding 1 Index          (b) Holding 2 Indices

(c) Holding 3 Indices          (d) Holding 4 Indices

(e) Holding 7 Indices          (f) Holding 10 Indices

Figure 10: GSMI portfolio performance with different indices number holding. The GSMI portfolio performance analysis examines strategies with varying numbers of constituent indices. The investment strategy involves rebalancing every five trading days by selecting from the 20 available indices, with portfolio returns calculated as the average performance of the held indices. We visualize comparative results for portfolios holding 1, 2, 3, 4, 7, and 10 indices simultaneously. Additionally, the average return of all 20 indices (20-Index Average) is provided as a benchmark for performance evaluation.

### B.5 Naive Model Details

**Naive Model Setup**. We construct a Naive model that uses the most recent value of the input time series as the predicted value. To address potential issues arising from zero standard deviation in the time series, we incorporate a minimal random perturbation by adding Gaussian noise with a standard deviation of 0.001.

**Forecast Samples**. The Figure 11 illustrates the prediction samples of this method across different datasets. The yellow line represents the Naive prediction, which remains nearly constant and fails to capture the dynamic variations in the time series. However, for time series without significant periodicity, the Naive model may achieve relatively small prediction errors. **Performance Comparison**. The Table 13 presents the MSE and MAE losses of the Naive model on the Exchange, ETTh2,

Table 13: Example for MSE&MAE comparison in four datasets. The red values indicate the best performance, while the blue values with an underscore represent the second-best performance.

| Dataset | Length | Naive | | iTransformer | | DLinear | | FEDformer | |
|---|---|---|---|---|---|---|---|---|---|
| | | MSE | MAE | MSE | MAE | MSE | MAE | MSE | MAE |
| Exchange | 96 | 0.081 | 0.196 | 0.086 | 0.206 | 0.088 | 0.218 | 0.148 | 0.278 |
| | 192 | 0.167 | 0.289 | 0.177 | 0.299 | 0.176 | 0.315 | 0.271 | 0.315 |
| | 336 | 0.306 | 0.398 | 0.331 | 0.417 | 0.313 | 0.427 | 0.460 | 0.427 |
| | 720 | 0.810 | 0.676 | 0.847 | 0.691 | 0.839 | 0.695 | 1.195 | 0.695 |
| ETTh2 | 96 | 0.281 | 0.369 | 0.297 | 0.349 | 0.333 | 0.387 | 0.358 | 0.397 |
| | 192 | 0.339 | 0.404 | 0.380 | 0.400 | 0.477 | 0.476 | 0.429 | 0.439 |
| | 336 | 0.372 | 0.424 | 0.428 | 0.432 | 0.594 | 0.541 | 0.496 | 0.487 |
| | 720 | 0.439 | 0.463 | 0.427 | 0.445 | 0.831 | 0.657 | 0.463 | 0.474 |
| ETTm2 | 96 | 0.202 | 0.310 | 0.180 | 0.264 | 0.193 | 0.292 | 0.203 | 0.287 |
| | 192 | 0.235 | 0.337 | 0.250 | 0.309 | 0.284 | 0.362 | 0.269 | 0.328 |
| | 336 | 0.270 | 0.361 | 0.311 | 0.348 | 0.369 | 0.427 | 0.325 | 0.366 |
| | 720 | 0.335 | 0.401 | 0.412 | 0.407 | 0.554 | 0.522 | 0.421 | 0.415 |
| Traffic | 96 | 2.715 | 1.077 | 0.395 | 0.268 | 0.650 | 0.396 | 0.587 | 0.366 |
| | 192 | 2.747 | 1.085 | 0.417 | 0.276 | 0.598 | 0.370 | 0.604 | 0.373 |
| | 336 | 2.789 | 1.094 | 0.433 | 0.283 | 0.605 | 0.373 | 0.621 | 0.383 |
| | 720 | 2.810 | 1.097 | 0.467 | 0.302 | 0.645 | 0.394 | 0.626 | 0.382 |

ETTm2, and Traffic datasets. For comparative analysis, the results of three other time series models (iTransformer, DLinear, and FEDformer results from [29]) are also included. The results suggest that the Naive model performs poorly on time series with strong periodicity, such as Traffic, but achieves smaller prediction errors on datasets like Exchange. This further emphasizes that prediction error alone is insufficient to comprehensively evaluate the forecasting capability of time series models.

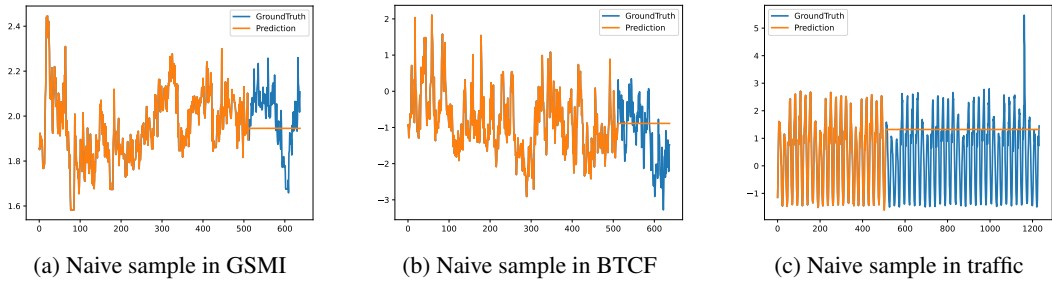

(a) Naive sample in GSMI     (b) Naive sample in BTCF     (c) Naive sample in traffic

Figure 11: Naive prediction samples. Naive model use the most recent value of the input time series as the predicted value, the prediction curve is represented by the straight-line segment of the yellow curve in the figure, as illustrated.

## C    Strategies Details

### C.1    Strategy Setup

Since the model predicts the price movement over the next few days, the final change in the predicted price is calculated as the Original Difference Signal. However, as shown in Figure 12a, this signal exhibits strong autocorrelation, meaning it does not remain stable over longer time horizons. This makes it unsuitable to use a single threshold as a timing condition, as it would fail to capture short-term trading signals effectively. To address this, a rolling average is applied to the signal time series, and the difference between the original signal and the rolling average is computed (as Difference in Difference). The resulting signal sequence, illustrated in Figure 12b, represents the extent to which the signal strength deviates from its recent average performance. The rolling window is set to one quarter, corresponding to approximately 63 trading days for tasks in the GSMI dataset, and set to 21 for tasks in the BTCF dataset.

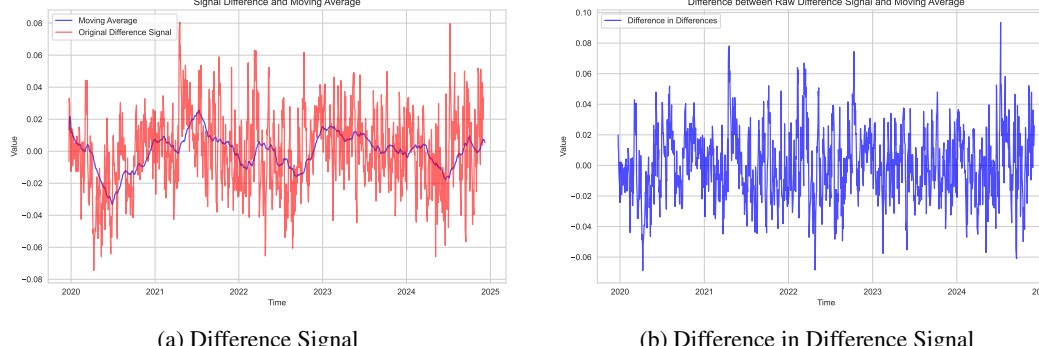

(a) Difference Signal          (b) Difference in Difference Signal

Figure 12: Difference in Difference. The Difference Signal in left exhibits autocorrelation, making it unsuitable for short-term trading signals. To address this, a rolling average is applied to create the Difference in Difference Signal (right), which reflects how much the signal deviates from its recent average performance. The rolling window is set to 63 for tasks in the GSMI dataset and 21 for tasks in the BTCF dataset. When the Difference in Difference signal is positive, a buy action is executed, and when negative, the strategy switches to cash or holds a short position. For portfolio optimization, indices are ranked by signal strength, and the strongest indices are selected based on the number of positions held.

When the Difference in Difference signal is positive, we execute a buy action. When the signal is negative, we execute either switching to cash or holding a short position. For portfolio optimization, at each rebalancing point, the signal strengths of different indices are ranked from highest to lowest. The strategy then selects indices with the highest signal strengths, depending on the number of indices being held.

## C.2 Explanations of Evaluation Metrics

**Annual Return**: The annual return is the average return an investment yields over the course of a year, often expressed as a compounded rate.

$$\text{Annual Return} = \left( \prod_{t=1}^{T} (1 + r_t) \right)^{\frac{1}{T}} - 1,$$

where:

- $r_t$ is the return at time $t$,

- $T$ is the total number of time periods (typically in years).

**Cumulative Returns**: The cumulative return represents the total return from the start to a specific point in time, taking into account all intermediate returns.

$$\text{Cumulative Return} = \prod_{t=1}^{T} (1 + r_t) - 1,$$

where:

- $r_t$ is the return at time $t$,

- $T$ is the total number of time periods.

**Annual Volatility**: The annual volatility is a measure of the variability or dispersion of returns, commonly used as a risk indicator.

$$\text{Annual Volatility} = \sqrt{252} \times \text{Std}(r),$$

where:

- $\text{Std}(r)$ is the standard deviation of the daily returns,

- 252 is the number of trading days in a year.

**Sharpe Ratio**: The Sharpe ratio is used to evaluate the return of an investment compared to its risk, measuring the excess return per unit of risk.

$$\text{Sharpe Ratio} = \frac{\text{Annual Return} - r_f}{\text{Annual Volatility}},$$

where:

- $r_f$ is the risk-free rate.

**Calmar Ratio**: The Calmar ratio measures the relationship between the annual return and the maximum drawdown, used to evaluate the risk-adjusted performance.

$$\text{Calmar Ratio} = \frac{\text{Annual Return}}{\text{Max Drawdown}},$$

where:

- The maximum drawdown is the largest peak-to-trough loss.

**Stability**: The stability of a strategy's returns is measured by the R-squared value of the linear fit to the cumulative log returns. A higher R-squared indicates more stable returns, while a lower value suggests volatility or inconsistency.

$$R^2 = \frac{\sum_{t=1}^{T}(\hat{y}_t - \bar{y})^2}{\sum_{t=1}^{T}(y_t - \bar{y})^2},$$

where:

- $\hat{y}_t$ are the predicted cumulative log returns from the linear regression,

- $y_t$ are the actual cumulative log returns,

- $\bar{y}$ is the mean of the actual cumulative log returns.

**Max Drawdown**: The maximum drawdown measures the largest percentage decline from the peak to the trough during a specific time period.

$$\text{Max Drawdown} = \min_{t \in [1,T]} \frac{P_t - P_{\max}}{P_{\max}},$$

where:

- $P_t$ is the portfolio value at time $t$,

- $P_{\max}$ is the maximum portfolio value up to time $t$.

**Omega Ratio**: The Omega ratio compares the probability-weighted gains to the probability-weighted losses, quantifying the return-to-risk tradeoff.

$$\text{Omega Ratio} = \frac{\sum_{r_i > 0} r_i}{\sum_{r_i < 0} |r_i|},$$

where:

- $r_i$ is the return at time $i$,

- Positive and negative returns are separated for the calculation.

**Sortino Ratio**: The Sortino ratio is similar to the Sharpe ratio but only considers downside risk (negative returns) in its calculation.

$$\text{Sortino Ratio} = \frac{\text{Annual Return} - r_f}{\text{Downside Volatility}},$$

where:

- Downside Volatility is the standard deviation of negative returns.

## D   Analysis of Error Metrics

In the financial sector, identifying significant tail signals, such as large upward or downward price fluctuations, is of great importance, the underlying distribution of the true values in such cases often does not follow a normal distribution[51, 2]. However, error metrics like MSE typically assume a Gaussian distribution, which means that for samples that do not meet the Gaussian distribution assumption, using MSE for evaluation will introduce systematic bias[50].

Specifically, when the error follows a normal distribution, let's assume the prediction error follows $r - \hat{r} = \epsilon \sim \mathcal{N}(0, \sigma^2)$, which means,

$$\mathcal{P}(r_i - \hat{r}_i) = \frac{1}{\sqrt{2\pi\sigma^2}} e^{-\frac{(r_i - \hat{r}_i)^2}{2\sigma^2}},$$

For $n$ independent and identically distributed samples, the joint likelihood function becomes

$$\mathcal{P}(\mathbf{r} - \hat{\mathbf{r}}) = \prod_{i=1}^{n} \mathcal{P}(r_i - \hat{r}_i) = \left(2\pi\sigma^2\right)^{-\frac{n}{2}} e^{-\frac{1}{2\sigma^2} \sum_{i=1}^{n}(r_i - \hat{r}_i)^2},$$

Minimizing the negative logarithm of the likelihood gives:

$$-\log \mathcal{P}(\mathbf{r}|\hat{\mathbf{r}}) = \frac{n}{2} \log(2\pi\sigma^2) + \frac{1}{2\sigma^2} \sum_{i=1}^{n}(r_i - \hat{r}_i)^2.$$

This is equivalent to minimizing the mean squared error:

$$\text{MSE} = \frac{1}{n} \sum_{i=1}^{n}(r_i - \hat{r}_i)^2.$$

However, the presence of sample errors following a mixture of Gaussian distributions can stem from various underlying factors. For example, when the true values are generated from multiple subpopulations, each characterized by distinct Gaussian distributions with potentially different biases. Additionally, when models are optimized using MSE as the loss function, their limited capacity to capture intricate data patterns may lead to residuals that inherently manifest as a mixture of Gaussian distributions.

Suppose the errors follow a Gaussian mixture model composed of two Gaussian distributions (as shown in Figure 13):

$$r - \hat{r} = \epsilon \sim \pi \mathcal{N}(\mu_1, \sigma_1^2) + (1 - \pi)\mathcal{N}(\mu_2, \sigma_2^2)$$

where $\pi \in (0, 1)$ is the mixture weight, $\mu_1$ and $\mu_2$ are the means of the two Gaussian components, one may positive and the other negative. $\sigma_1^2$ and $\sigma_2^2$ are their respective variances

In this case, the error probability density function will exhibit two local maxima (when $\mu_1 \neq \mu_2$), centered at $\mu_1$ and $\mu_2$ respectively. Minimizing the negative log-likelihood then yields:

$$-\log \mathcal{P}(r - \hat{r}) \propto \sum_{i=1}^{n} \log \left[ \pi e^{-\frac{(r_i - \hat{r}_i - \mu_1)^2}{2\sigma_1^2}} + (1 - \pi)e^{-\frac{(r_i - \hat{r}_i - \mu_2)^2}{2\sigma_2^2}} \right]$$

This formulation is not equivalent to minimizing the MSE. The key difference lies in the mixture model's ability to account for multiple error regimes, whereas MSE implicitly assumes a single Gaussian error distribution.

When we take the derivative of MSE with respect to $\hat{r}$ and set it to zero

$$\frac{\partial}{\partial \hat{r}} E[(r - \hat{r})^2] = -2E[r - \hat{r}] = 0,$$

we obtain the optimal predictor

$$\hat{r}^*_{MSE} = E[r] = E[\hat{r} + \epsilon] = \hat{r} + E[\epsilon]$$

Given that the expected error is

$$E[\epsilon] = \pi\mu_1 + (1 - \pi)\mu_2$$

the final optimal prediction becomes

$$\hat{r}^* = \hat{r} + \pi\mu_1 + (1 - \pi)\mu_2$$

The MSE criterion exhibits a tendency to "average out" the bimodal characteristics of the mixture distribution, introducing a systematic bias in estimation. This phenomenon occurs because MSE optimization inherently favors predictions toward the distribution's mean rather than its actual modes. When dealing with multimodal conditional distributions $P(r \mid x)$ (where $x$ represents conditioning information), the single-point estimates $\hat{r}$ produced by conventional models are forced to compromise between competing modes.

This creates an evaluation paradox where predictions close to either mode (but distant from the mean) are penalized disproportionately, while predictions near the arithmetic mean (potentially far from all genuine modes) receive favorable MSE scores. A concrete example would be predicting $\frac{a+b}{2}$ for a true distribution concentrated at values $a$ and $b$ - such predictions minimize MSE while failing to capture the actual bimodal nature of the data generating process.

Besides, the correlation coefficient can serve as a complementary evaluation metric to error measures. In mixture distributions, the Pearson correlation coefficient remains effective at revealing the predictive relationship between forecasts and actual values. Specifically, for any distribution, the MSE can be decomposed as follows:

$$
\begin{aligned}
\text{MSE} &= \mathbb{E}\left[(r - \hat{r})^2\right] \\
&= \mathbb{E}\left[r^2\right] - 2\mathbb{E}\left[r\hat{r}\right] + \mathbb{E}\left[\hat{r}^2\right] \\
&= (\mu_r - \mu_{\hat{r}})^2 + \text{Var}(r) + \text{Var}(\hat{r}) - 2\text{Cov}(r, \hat{r}) \\
&= (\mu_r - \mu_{\hat{r}})^2 + (\sigma_r - \sigma_{\hat{r}})^2 + 2\sigma_r\sigma_{\hat{r}}(1 - \rho)
\end{aligned}
$$

where $\rho$ represents the Pearson correlation coefficient. The MSE decomposition reveals three distinct components. 1.The squared bias term $(\mu_r - \mu_{\hat{r}})^2$ captures systematic deviation between predictions and true values. 2.The variance mismatch $(\sigma_r - \sigma_{\hat{r}})^2$ quantifies scale misalignment. 3.The covariance loss $2\sigma_r\sigma_{\hat{r}}(1 - \rho)$ measures errors from trend inconsistency.

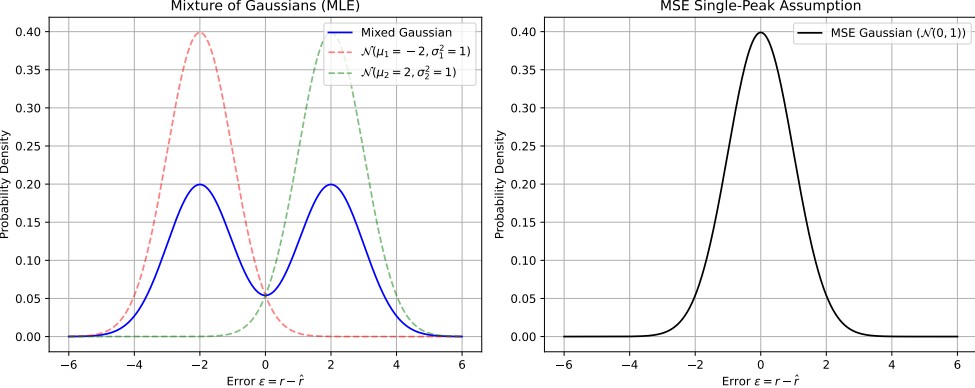

Figure 13: Comparison of error distributions under Maximum Likelihood Estimation versus MSE. Left: Mixture of Gaussians distribution with bimodal peaks at $\mu_1$ and $\mu_2$, shows the ability of MLE to capture error distribution even on multimodal structures. Right: Unimodal Gaussian assumption (centered at zero) for error distribution, both MLE and MSE works fine on this.

Under the Gaussian mixture assumption, the first two components may introduce systematic errors. When these biases become sufficiently large, they can dominate the MSE loss, potentially obscuring the underlying correlation structure. This occurs because the bias and variance terms reflect distribution-level mismatches that may overwhelm the correlation component in the overall error evaluation.

# E Limitations

Although we have explored building bridges between cutting-edge time-series models and financial time-series applications, our study has several limitations:

1. The three financial datasets we constructed do not encompass all types of financial time series. In particular, the high-frequency OPTION dataset (with 1-minute sampling frequency) covers only one year of data, which may fail to include longer-term temporal patterns.

2. While our portfolio strategy reduces risk through diversification across multiple assets, it does not incorporate model-based risk prediction and allocation. Additionally, transaction costs (e.g., commissions) were ignored during backtesting, though they are crucial in practical trading environments.

3. We did not develop new time-series models, nor did we include pre-trained large models as benchmarks in our evaluation.

