# OpenReview forum: "FinTSBridge: A New Evaluation Suite for Real-world Financial Prediction with Advanced Time Series Models"
_NeurIPS.cc/2025/Datasets_and_Benchmarks_Track — Submitted to NeurIPS 2025 Datasets and Benchmarks Track_

### Official Review · Reviewer_s5TB · 2025-06-26

**Rating:** 5
**Confidence:** 4

**Summary:**

This study addresses the gap between time series forecasting models and their application in financial asset pricing. The authors construct three financial datasets, evaluate over ten recent forecasting models, and introduce two new metrics (msIC and msIR) to assess time series correlations. Through tailored task designs, they demonstrate that with domain-specific adaptations, these models perform effectively across various asset classes. The proposed evaluation framework, FinTSBridge, aims to facilitate robust assessment of forecasting models in financial contexts.

**Additional Feedback:**

N.A.

**Dataset Code Accessibility:**

Yes

**Dataset Code Comments:**

Dataset is provided on Kaggle and code is available on anonymous repo.

**Ethical Comments:**

N.A.

**Ethical Considerations:**

No, there are no or only very minor ethics concerns

**Final Justification:**

The dataset released in this paper is a valuable complement to existing time series evaluation benchmarks. During the rebuttal phase, the authors conducted additional experiments using recent time series foundation models, which strengthened the technical contribution of the paper.

**Limitations Weaknesses:**

1. Recent time series foundation models—such as TimesFM-2.0, Moirai-MoE, and Chronos-Bolt—are not cited or discussed in the related work section. Their omission limits the contextual positioning of this paper within the latest literature.
2. In Appendix C, the authors describe the strategy setup and evaluation metrics, but no corresponding results are reported. Could you please clarify on this point?
3. While the overall writing is solid, some important experiments (those claimed in the introduction part, e.g., timing trading) has been deferred to the appendix. The authors may consider reducing the length of the related work section and moving some of it to the appendix to make room for key content in the main paper.

**Strengths Contributions:**

1. The writing is clear and coherent. The motivation for constructing the dataset is well-articulated and justified.
2. The released datasets are a good complement to the existing time series evaluation benchmarks (e.g. ETT series, transportation, electricity, illness, etc.), and thus may have a good impact on the field of time series forecasting.
3. The dataset is benchmarked using more than ten state-of-the-art time series forecasting models, providing a comprehensive performance comparison.
4. The authors apply data preprocessing to address distribution shift, introduce new evaluation metrics, and design financial-specific tasks such as portfolio optimization, which adds practical relevance to the study.

---

> ### Author Rebuttal · Authors · 2025-07-31
>
> First of all, Thank you for your valuable time and thoughtful feedback. We are deeply grateful for your suggestion of adding three time series foundation models as benchmarks, which addresses the limitation that the current benchmark models in this work do not cover foundation models. We also thank you for proposing suggestions on associating and adjusting the context of important experiments and key content in the paper, and we will follow these suggestions in the revision.
>
> We will now address your concerns and questions one by one:
>
> **W1: Discussion on the latest time series foundation models**
>
> We agree that incorporating the latest time series foundation models will better enhance this work. We greatly appreciate your suggestion of including the latest foundation models, such as TimesFM-2.0, Moirai-MoE, and Chronos-Bolt. We have conducted a detailed investigation into these foundation models and conduct experimental evaluations.
>
> Specifically, we selected the publicly released versions with the largest parameter sizes for each model (as the table below). Each model take multiple rounds of fine-tuning for different tasks, and each experiment was run three times with different random seeds to ensure robust model performance. For tasks with a prediction length of 5, we have provided complete evaluation results for all three models. For TimesFM-2.0 and Chronos-Bolt, we have also reported evaluation data for other prediction lengths. Moving forward, we will supplement the fine-tuning evaluation data for all prediction lengths in the revised version.
>
> |            | TimesFM-2.0 | Moirai-MoE (Base) | Chronos-Bolt (Base) |
> |------------|-------------|-------------------|---------------------|
> | Model size | 500M params | 935M params       | 205M params         |
>
> The results of multivariate-to-multivariate prediction are shown in the table below. From the experimental results, it can be seen that compared with small models, large models generally achieve more competitive results after fine-tuning. We have also provided the experimental results of multivariate-to-univariate prediction, which are included in the response to Reviewer BA5H's W2. These results similarly indicate that large models perform competitively in multivariate-to-univariate prediction tasks.
>
> **Results of multivariate-to-multivariate**
>
> | Method       | Metric |      GSMI      |               |               |               |     OPTION    |               |               |               |      BTCF     |               |               |               |
> |--------------|--------|:--------------:|:-------------:|:-------------:|:-------------:|:-------------:|:-------------:|:-------------:|:-------------:|:-------------:|:-------------:|:-------------:|:-------------:|
> |              |        | 5              | 21            | 63            | 126           | 5             | 21            | 63            | 126           | 5             | 21            | 63            | 126           |
> | TimesFM-2.0  | MSE    | 0.0582±0.0004  | 0.0887±0.0024 | 0.1425±0.0016 | 0.1848±0.0058 | 0.1809±0.0054 | 0.2323±0.0027 | 0.3008±0.0071 | 0.3675±0.0059 | 0.1611±0.0025 | 0.1936±0.0031 | 0.2136±0.0018 | 0.2255±0.0017 |
> |              | MAE    | 0.1045±0.0011  | 0.1553±0.0028 | 0.2250±0.0005 | 0.2783±0.0033 | 0.1734±0.0035 | 0.2076±0.0011 | 0.2535±0.0016 | 0.2962±0.0016 | 0.1869±0.0017 | 0.2107±0.0016 | 0.2314±0.0012 | 0.2493±0.0013 |
> |              | msIC   | -0.0023±0.0119 | 0.0319±0.0026 | 0.0284±0.0071 | 0.0575±0.0133 | 0.0420±0.0210 | 0.0590±0.0040 | 0.0337±0.0068 | 0.0412±0.0040 | 0.1039±0.0042 | 0.1631±0.0024 | 0.1912±0.0103 | 0.2249±0.0025 |
> |              | msIR   | 0.0039±0.0185  | 0.0914±0.0067 | 0.1118±0.0105 | 0.2034±0.0274 | 0.0852±0.0541 | 0.1501±0.0103 | 0.0900±0.0161 | 0.1144±0.0107 | 0.1969±0.0083 | 0.5201±0.0040 | 0.9882±0.0181 | 1.4087±0.0204 |
> | Chronos-Bolt | MSE    | 0.0587±0.0002  | 0.0877±0.0006 | 0.1354±0.0006 |       -       | 0.1810±0.0013 | 0.2180±0.0003 | 0.2733±0.0007 |       -       | 0.1468±0.0007 | 0.1661±0.0007 | 0.1814±0.0006 |       -       |
> |              | MAE    | 0.1027±0.0000  | 0.1522±0.0030 | 0.2190±0.0010 |       -       | 0.1625±0.0004 | 0.1932±0.0003 | 0.2366±0.0004 |       -       | 0.1728±0.0003 | 0.1932±0.0001 | 0.2154±0.0005 |       -       |
> |              | msIC   | 0.0399±0.0002  | 0.0334±0.0051 | 0.0384±0.0023 |       -       | 0.0687±0.0053 | 0.0920±0.0050 | 0.0931±0.0025 |       -       | 0.1213±0.0032 | 0.1938±0.0036 | 0.2255±0.0027 |       -       |
> |              | msIR   | 0.0652±0.0004  | 0.0780±0.0044 | 0.0958±0.0035 |       -       | 0.1230±0.0112 | 0.3116±0.0468 | 0.4186±0.0721 |       -       | 0.2259±0.0049 | 0.6446±0.0065 | 1.2727±0.0067 |       -       |
> | Moirai-MoE   | MSE    | 0.0566±0.0004  |               |               |               | 0.1781±0.0011 |               |               |               | 0.1550±0.0012 |               |               |               |
> |              | MAE    | 0.1030±0.0002  |               |               |               | 0.1716±0.0002 |               |               |               | 0.1830±0.0006 |               |               |               |
> |              | msIC   | 0.0412±0.0004  |               |               |               | 0.0608±0.0008 |               |               |               | 0.1062±0.0013 |               |               |               |
> |              | msIR   | 0.0710±0.0003  |               |               |               | 0.1622±0.0005 |               |               |               | 0.2004±0.0009 |               |               |               |
>
> **W2: Explanation of the result reported for Appendix C**
>
> We agree that we should add redirects in Appendix C to enhance relevance. In fact, we have reported this result in Table 2 of the main text and set an index redirecting to Appendix C. However, we should also add an index in Appendix C that links back to Table 2 of the main text to strengthen content relevance, and we will supplement this in the revised version.
>
>
> **W3: Reasonable adjustment of content in the main text and appendix**
>
> We agree that reasonably moving some parts of the related work to the appendix and placing key results (such as timing trading results and empirical experimental results related to the introduction section) in the main text will better facilitate the overall expression. We should consider the perspective of readers more to better connect the content layout of the work. We will follow this principle to adjust the key content.
>
> We would like to express our sincere gratitude again for your review and specific suggestions, which enable us to further improve our paper.

---

> > ### Comment · Reviewer_s5TB · 2025-08-03
> >
> > Dear authors, thank you for your rebuttal and for providing more experimental results. Regarding Moirai-MoE, I am wondering why some results are missing from the rebuttal. Thank you.

---

> > ### Comment · Reviewer_s5TB · 2025-08-05
> >
> > One more suggestion I would make is that, instead of fine-tuning TimesFM-2.0, Moirai-MoE, and Chronos-Bolt, since they are time series foundation models, it would also be valuable to demonstrate their zero-shot performance on the proposed financial datasets (which can be quickly tested and implemented than fine-tuning), and compare these zero-shot results with the full-shot deep learning methods used in this paper. These results can clearly show how the current foundation models handle these new datasets.

---

> > > ### Author Response · Authors · 2025-08-05
> > >
> > > We are sincerely thank you for your valuable suggestions. We agree that we should add performance comparison of zero-shot experiments for time series foundation models. We plan to include the zero-shot results of these foundation models in the revised version and compare them with full-shot deep learning methods. This kind of comparison is likely more suitable for real-world application scenarios, and we are also very interested in it. Finally, we are deeply grateful for your suggestions, as these precious insights have made our paper more meaningful!

---

### Official Review · Reviewer_BA5H · 2025-07-02

**Rating:** 5
**Confidence:** 3

**Summary:**

Briefly, this paper presents a framework to bridge the gap between advanced time series forecasting models and real-world financial applications. The authors curate three financial datasets (GSMI, OPTION, BTCF), propose new evaluation metrics (msIC and msIR), and conduct extensive experiments with over ten state-of-the-art time series models. The work aims to provide actionable insights for financial decision-making by emphasizing correlation-aware forecasting and practical financial tasks.

**Additional Feedback:**

This paper presents a valuable contribution to financial time series forecasting by introducing new datasets, metrics, and a comprehensive evaluation framework. The three datasets cover different financial instruments and frequencies, addressing a gap in existing benchmarks. Besides, the pre-processing steps are well-documented and justified. Besides, the proposed evaluation metrics move beyond traditional error measures by incorporating temporal correlation and robustness.

However, its weaknesses, particularly in dataset diversity and model coverage, lead to the current rating.

**Dataset Code Accessibility:**

Yes

**Ethical Considerations:**

No, there are no or only very minor ethics concerns

**Final Justification:**

This paper presents a valuable contribution to financial time series forecasting by introducing new datasets, metrics, and a comprehensive evaluation framework. The three datasets cover different financial instruments and frequencies, addressing a gap in existing benchmarks. During the rebuttal period, the authors have addressed all of my concerns.

**Limitations Weaknesses:**

+ The diversity of the three datasets is limited. While the three datasets cover different financial instruments (indices, options, crypto), they are still narrow in scope. For example, the OPTION dataset only includes CSI 300 ETF options from the Chinese market, which may limit generalizability to other derivatives or markets. Similarly, the BTCF dataset focuses solely on Bitcoin, omitting other cryptocurrencies.

+ The paper evaluates a wide range of models but does not justify the exclusion of other relevant architectures (e.g., recent LLM-based time series models).

+ The msIC and msIR metrics are empirically motivated but lack theoretical grounding. Besides, the backtesting of trading strategies assumes zero transaction costs, which is unrealistic. In high-frequency or crypto markets, transaction costs (e.g., fees, slippage) can significantly impact net returns.

**Strengths Contributions:**

+ This paper introduces three well-curated financial datasets covering diverse asset classes and frequencies, which fill a gap in existing benchmarks. The new metrics (msIC and msIR) are thoughtfully designed to capture temporal correlation and robustness, addressing limitations of traditional error metrics like MSE and MAE.

+ This paper rigorously evaluates 16 time series models across multiple forecasting tasks (multivariate-to-multivariate, univariate, and partial forecasting) and provides detailed performance comparisons.

+ The preprocessing techniques for financial time series are well-justified, and the paper provides clear explanations of dataset construction, metric design, and experimental setups. The results are supported by extensive tables and visualizations.

---

> ### Author Rebuttal · Authors · 2025-07-31
>
> First of all, we would like to thank you for the valuable time and thoughtful considerations. We appreciate your suggestions, especially for expanding the types of models and the coverage of financial data, which will help enhance the generalizability of our work. We accept these suggestions and will follow them in the revision.
>
> Next, we will address your concerns and questions one by one:
>
> **W1: Enhancing diversity of datasets**
>
> We agree with this point. Although we have made efforts to include a more diverse range of financial assets, feature indicators, and sampling frequencies, there are still many assets not included in the dataset. We will continue to open up a broader range of financial datasets in the future, including more digital currencies and derivatives, and maintain more model performance evaluations based on these expanded datasets.
>
>
> **W2: Time series large models should not be excluded from the benchmarks**
>
> We agree that time series large models should be included in the benchmarks. We have added recent time series foundation models, including TimesFM-2.0, Moirai-MoE, and Chronos-Bolt, and tested their performance on the datasets. To reasonably manage the length of the response, we have documented the experimental results of multivariate-to-multivariate prediction in the reply to Reviewer s5TB's W1, and we report the experimental results of multivariate-to-univariate prediction in the table below.
>
> Given the constraints of time and computing power, we have obtained evaluation results for most foundation models. From the reported results, it appears that these foundation models, after several rounds of fine-tuning, show highly competitive predictive performance in both multivariate-to-univariate and multivariate-to-multivariate prediction experiments. Including these large models in our evaluations helps supplement the diversity of the benchmarks. We will complete all fine-tuning evaluation results in the revised version to provide a more comprehensive overview of performance across different model types.
>
> **Results of multivariate-to-univariate**
>
> | Method       | Metric |      GSMI     |               |               |               |     OPTION    |               |               |               |      BTCF      |               |               |               |
> |--------------|--------|:-------------:|:-------------:|:-------------:|:-------------:|:-------------:|:-------------:|:-------------:|:-------------:|:--------------:|:-------------:|:-------------:|:-------------:|
> |              |        | 5             | 21            | 63            | 126           | 5             | 21            | 63            | 126           | 5              | 21            | 63            | 126           |
> | TimesFM-2.0  | MSE    | 0.0089±0.0004 | 0.0286±0.0003 | 0.0582±0.0001 | 0.0971±0.0026 | 0.3466±0.0032 | 0.4365±0.0060 | 0.5570±0.0025 | 0.7910±0.0459 | 0.0003±0.0000  | 0.0010±0.0000 | 0.0029±0.0000 | 0.0055±0.0000 |
> |              | MAE    | 0.0651±0.0035 | 0.1161±0.0002 | 0.1677±0.0020 | 0.2323±0.0010 | 0.4170±0.0040 | 0.4661±0.0028 | 0.5228±0.0015 | 0.6233±0.0211 | 0.0112±0.0005  | 0.0214±0.0001 | 0.0392±0.0002 | 0.0540±0.0003 |
> |              | msIC   | 0.0642±0.0091 | 0.0602±0.0072 | 0.1825±0.0855 | 0.3382±0.0080 | 0.0523±0.0312 | 0.0791±0.0167 | 0.0486±0.0008 | 0.0286±0.0033 | -0.0064±0.0035 | 0.0076±0.0107 | 0.0642±0.0068 | 0.0993±0.0034 |
> |              | msIR   | 0.1006±0.0151 | 0.1076±0.0130 | 0.3417±0.1751 | 0.6245±0.0063 | 0.0920±0.0574 | 0.2166±0.0400 | 0.1787±0.0031 | 0.1005±0.0088 | -0.0106±0.0058 | 0.0138±0.0192 | 0.1226±0.0095 | 0.1883±0.0045 |
> | Chronos-Bolt | MSE    | 0.0075±0.0000 | 0.0270±0.0005 | 0.0606±0.0001 |       -       | 0.3169±0.0021 | 0.4212±0.0005 | 0.5323±0.0026 |       -       | 0.0002±0.0000  | 0.0010±0.0000 | 0.0032±0.0000 |       -       |
> |              | MAE    | 0.0578±0.0025 | 0.1071±0.0013 | 0.1763±0.0003 |       -       | 0.3904±0.0009 | 0.4541±0.0009 | 0.5127±0.0025 |       -       | 0.0099±0.0000  | 0.0221±0.0002 | 0.0407±0.0003 |       -       |
> |              | msIC   | 0.0827±0.0386 | 0.1072±0.0077 | 0.1622±0.0028 |       -       | 0.1280±0.0001 | 0.1205±0.0004 | 0.0774±0.0034 |       -       | 0.0027±0.0109  | 0.0230±0.0012 | 0.0444±0.0020 |       -       |
> |              | msIR   | 0.1224±0.0560 | 0.1632±0.0123 | 0.2948±0.0036 |       -       | 0.2118±0.0007 | 0.3066±0.0014 | 0.2565±0.0145 |       -       | 0.0043±0.0174  | 0.0392±0.0019 | 0.0784±0.0043 |       -       |
> | Moirai-MoE   | MSE    | 0.0071±0.0002 |               |               |               | 0.3331±0.0030 |               |               |               | 0.0002±0.0000  |               |               |               |
> |              | MAE    | 0.0591±0.0007 |               |               |               | 0.4015±0.0013 |               |               |               | 0.0099±0.0002  |               |               |               |
> |              | msIC   | 0.0050±0.0053 |               |               |               | 0.1100±0.0041 |               |               |               | 0.0036±0.0188  |               |               |               |
> |              | msIR   | 0.0076±0.0081 |               |               |               | 0.1889±0.0115 |               |               |               | 0.0062±0.0300  |               |               |               |
>
>
> **W3: Lack of theoretical basis for the msIC and msIR metrics; additionally, transaction cost impacts should be considered in trading strategy backtests**
>
> We agree with these points. The msIC and msIR metrics are empirically motivated. They address the limitation that the Information Coefficient and Information Ratio metrics cannot be applied to cross-multivariate multi-step prediction conditions, while establishing a connection between the multi-step predictions of time series models and financial data. These metrics provide evaluation information from the dimension of temporal correlation and serve as a supplement to error-based methods. In Appendix D, we analyze the limitations of error metrics and the supplementary role of correlation-based methods.
>
> Regarding transaction costs, we agree that they have a significant impact on net returns. However, the backtests in this work are primarily intended to clearly demonstrate the relative comparison results among benchmarks. Additionally, the choice of transaction fee rates can be influenced by numerous factors, including policy changes, transaction size, broker selection, and strategy optimization, etc. Our initial intention was to minimize the introduction of external variables.
>
> We understand the importance of considering transaction costs. Therefore, we have compiled some fee rate information, including the current fee schedule of Binance as shown in the table below. From the perspective, using USDC for U-based contract trading in maker mode is currently a favorable option. For index trading strategies, we believe they should be used in conjunction with stock strategies or applied to index ETF trading. In some markets, the transaction fees for index ETFs are relatively low, approximately 0.005% to 0.03%, while those for ETF options are around 0.01%. Readers concerned with transaction costs can estimate the approximate transaction costs by multiplying the fee rate by the strategy’s level of turnover rate. We hope this information will fill the gap in the discussion of transaction costs in the revised version.
>
> | Level        | Spot Maker/Taker (%)  | Futures USDT Maker/Taker (%) | Futures USDC Maker/Taker (%) |
> |--------------|-----------------------|------------------------------|------------------------------|
> | Regular User | 0.1 / 0.1             | 0.02/ 0.05                   | 0/ 0.04                      |
> | VIP 1        | 0.09/ 0.1             | 0.016/ 0.04                  | 0/ 0.032                     |
> | VIP 2        | 0.08/ 0.1             | 0.014/0.035                  | 0/ 0.028                     |
> | VIP 3        | 0.04/ 0.06            | 0.012/0.032                  | 0/ 0.0256                    |
> | VIP 4        | 0.04/ 0.052           | 0.01/ 0.03                   | 0/ 0.024                     |
> | VIP 5        | 0.025/ 0.031          | 0.008/ 0.027                 | 0/ 0.0216                    |
> | VIP 6        | 0.02/ 0.029           | 0.006/ 0.025                 | 0/ 0.02                      |
> | VIP 7        | 0.019/ 0.028          | 0.004/ 0.022                 | 0/ 0.0176                    |
> | VIP 8        | 0.016/ 0.025          | 0.002/ 0.02                  | 0/ 0.016                     |
> | VIP 9        | 0.011/ 0.023          | 0/ 0.017                     | 0/ 0.0136                    |
>
> **F. Concerns about diversity of dataset and coverage of model**
>
> We agree with these concerns. As mentioned above, regarding model coverage, we have added several state-of-the-art time series foundation models as benchmarks. For dataset diversity, we will continue to expand the current dataset by releasing more high-quality financial datasets.
>
> We would like to sincerely thank you again for your review and specific suggestions, which enable us to further improve our work.

---

> > ### Comment · Reviewer_BA5H · 2025-08-04
> >
> > Thank you for the response. Most of my concerns have been addressed. I will increase my rating to 5 for acceptance.

---

> > > ### Author Response · Authors · 2025-08-04
> > >
> > > Thank you sincerely for your feedback and for considering our rebuttal in your assessment of the paper! Next, we will follow your advice to further explore the inclusion of more financial datasets and conduct additional model evaluations.

---

### Official Review · Reviewer_yZjs · 2025-07-02

**Rating:** 5
**Confidence:** 4

**Summary:**

This paper introduces three financial datasets spanning different time granularities and domains: GSMI (daily global indices), OPTION (minute-level CSI 300ETF options), and BTCF (hourly Bitcoin spot and futures). It further proposes a robust data preprocessing method and multi-step forecasting metrics tailored to financial tasks. Experiments on 16 time series models highlight gaps in the current financial forecasting area.

**Additional Feedback:**

F1. The article does not clearly define or distinguish the three prediction tasks, which hinders readers’ understanding of the rationale behind the subsequent investment experiments. Providing explicit definitions and highlighting the differences among these tasks would enhance the clarity and coherence of the experimental design.

F2. There are minor issues with the use of notations and references. Appendix B.1 contains a citation placeholder—“PSformer is from **(author?)**”—which lacks a proper bibliographic reference. Additionally, the title of Appendix B.4 is mislabelled and should be corrected to “Additional Multivariate-to-Partial Forecasting Results” instead of “Additional Multivariate-to-Univariate Forecasting Results.”

**Dataset Code Accessibility:**

Yes

**Dataset Code Comments:**

The CSV files for all three datasets are available at https://www.kaggle.com/datasets/timalex/fintsbridge or can be downloaded directly via the Kaggle API using the dataset identifier `timalex/fintsbridge`.

**Ethical Considerations:**

No, there are no or only very minor ethics concerns

**Final Justification:**

All my concerns have been addressed, thus I update my score from 4 to 5. This benchmark is important for the stock forecasting task.

**Limitations Weaknesses:**

W1. The effectiveness of the proposed metrics, such as msIC and msIR, is not fully supported by experimental results. For instance, DLinear and Stationary models perform poorly on these metrics in Tables 9 and 10 (GSMI dataset), yet achieve high cumulative returns in Figure 8. A similar inconsistency is observed with Koopa, which ranks second in cumulative returns in Figure 9 but shows subpar results on the BTCF dataset. These mismatches suggest a weak correlation between the proposed metrics and actual financial returns, raising concerns about the metrics' reliability in evaluating return-maximizing models.

W2. The article lacks a consistent visualization framework. In Appendix B, investment experiments are unevenly reported across forecasting tasks and datasets: Multivariate-to-Multivariate excludes investment evaluation, Multivariate-to-Univariate omits OPTION, and Multivariate-to-Partial includes only GSMI. This fragmented design, possibly due to varying time granularities, limits cross-dataset comparability and hinders reproducibility and future extensions.

W3. The article lacks a thorough review of recent advances in financial time series prediction. Prior studies [1–4] have explored diverse datasets (e.g., NASDAQ, NYSE, SP500, CSI300), tasks (e.g., stock trend and return prediction), and evaluation metrics (e.g., IC, ICIR, RIC), with well-designed investment experiments. However, the article neither compares its proposed datasets with existing benchmarks nor discusses differences in experimental design, limiting the clarity of its contributions and innovations.

[1] Li T, Liu Z, Shen Y, et al. Master: Market-guided stock transformer for stock price forecasting[C]//Proceedings of the AAAI Conference on Artificial Intelligence. 2024, 38(1): 162-170.

[2] Fan J, Shen Y. StockMixer: a simple yet strong MLP-based architecture for stock price forecasting[C]//Proceedings of the AAAI Conference on Artificial Intelligence. 2024, 38(8): 8389-8397.

[3] Qian H, Zhou H, Zhao Q, et al. Mdgnn: Multi-relational dynamic graph neural network for comprehensive and dynamic stock investment prediction[C]//Proceedings of the AAAI Conference on Artificial Intelligence. 2024, 38(13): 14642-14650.

[4] Xiang Q, Chen Z, Sun Q, et al. RSAP-DFM: regime-shifting adaptive posterior dynamic factor model for stock returns prediction[C]//Proceedings of the Thirty-Third International Joint Conference on Artificial Intelligence, IJCAI-24. 2024: 6116-6124.

**Strengths Contributions:**

S1.This paper addresses the fragmentation of existing time series forecasting tasks in the financial domain and introduces a novel dataset collection covering diverse financial sectors, markets, and time granularities.

S2. Within the proposed experimental framework, the study evaluates numerous state-of-the-art time series models and conducts extensive investment experiments.

S3.The article is clearly structured, with comprehensive experimental results and accessible mathematical derivations that enhance overall readability.

---

> ### Author Rebuttal · Authors · 2025-07-31
>
> First of all, we would like to thank you for taking the time to carefully review our paper. We believe that the reviewer’s suggestions—including a comprehensive review of the financial forecasting field, consistent visualization, explanations of indicator correlations, and revisions to symbol citations—will all help to further enhance the comprehensiveness and readability of the paper.
>
> We will now address the reviewer’s concerns and questions one by one:
>
> **W1: Discrepancies between correlation metrics and cumulative return metric in experimental results**
>
> We agree that in Table 9, models such as DLinear and Stationary perform relatively suboptimal in these metrics, yet they achieve relatively high cumulative returns in Figure 8. However, from the perspective of the evaluation metric formulas, msIC and msIR measure the overall average correlation level. In contrast, cumulative return metrics may be more susceptible to extreme samples. For instance, the SPX index price dropped by 35% from March to April 2020, and timing decisions during this period would have a significant impact on cumulative return metrics. Additionally, the evaluation results of the DLinear model show relatively large fluctuations across multiple experimental runs with different random seeds. This is reflected in larger standard deviation values of its correlation metrics and may thus lead to a lower mean of these metrics.
>
> **W2: Uneven reporting of investment experiments across prediction tasks and datasets**
>
> We agree with this observation. We have designed three categories of model evaluation experiments, which respectively predict all multivariate variables, partial multivariate variables, and univariate variables. The visualization of investment experiments is ultimately determined by combining the characteristics of datasets with the features of tasks. Indeed, it is challenging to construct consistent visualizations for all datasets. For instance, in the case of options data, implied volatility is often used as the prediction target, which inherently differs from the visualization of price predictions. Therefore, our original intention was to selectively adopt differentiated investment experiments and visualization methods based on task objectives in practical scenarios, with the premise of ensuring the completeness and scalability of the visualization methods.
>
>
> **W3: Lack of a thorough review of recent advances in financial time series prediction**
>
> We agree that a review of recent advances in this field should be supplemented. We have carefully examined studies [1-4] as well as some earlier related works. These studies constructed specialized network structures for financial time series prediction; however, they primarily focus on asset selection and generally adopt single-step forecasting. In terms of datasets, they typically concentrate on a single type of dataset—for example, mainly forecasting the returns of numerous stocks. In contrast, our work emphasizes integrating the multi-step time series forecasting capabilities of general time series prediction models with the inherent nature of financial time series forecasting, thereby establishing distinct forecasting schemes and evaluation methods that differ from previous approaches.
>
> Despite these differences, we agree that supplementing a comprehensive review of the latest advances in the field of financial time series prediction is necessary. This will help to further clarify the contributions and innovations of our work. We will add a review and comparison of relevant research in this area in the revised version, and we are also very grateful to the reviewer for providing these valuable references.
>
> **F1: The three prediction tasks need to be clearly defined or distinguished**
>
> We agree with this point. The three prediction tasks we constructed—multivariate-to-multivariate prediction, multivariate-to-partial-variable prediction, and multivariate-to-univariate prediction—require clearer differentiation in terms of their differences and application scenarios. For example, the scenario of multivariate-to-partial-variable prediction is particularly suitable for the GSMI dataset, where time series data of multiple global market indices exist, making it ideal for studying cross-market price movements and dependencies. We will enhance the differentiating statements of these tasks and clarify the motivations behind the experiments in the revised version.
>
> **F2. Issues with symbols and citations**
>
> Thank you for your careful reading, which has allowed us to identify and correct the citation placeholders and incorrect title labels in the appendix within the paper. We are deeply grateful for this!
>
> We would like to sincerely thank you again for your review and specific suggestions, which enable us to further improve our work.

---

> > ### Comment · Reviewer_yZjs · 2025-08-06
> >
> > Thank you so much for your rebuttal. Most of my concerns have been addressed, so I will update my final score to 5. Good Work!

---

> > > ### Author Response · Authors · 2025-08-06
> > >
> > > Thank you sincerely for your feedback and for reconsidering the assessment of our paper based on the discussions during the rebuttal! Moving forward, we will follow your careful review suggestions to complete the revised version of the paper.

---

### Official Review · Reviewer_CYiu · 2025-07-03

**Rating:** 5
**Confidence:** 1

**Summary:**

To deal with the challenges in real-world financial time series forecasting, this work proposes a framework called FinTSBridge, including three new datasets, two new metrics, and some financial-specific tasks.

Primary contributions:
1. Datasets: They constructed three new datasets—GSMI, OPTION, and BTCF—tailored to capture the nuances of financial asset pricing.
2. Metrics: Two novel metrics, msIC (mean sequential correlation) and msIR (correlation stability ratio), were developed to evaluate multi-variable multi-step predictions. These metrics address the limitations of traditional point-wise metrics (e.g., MSE, MAE) by focusing on temporal and sequential correlations.
3. Tasks: The paper introduces financial-specific tasks such as index portfolio optimization, timing trading, and BTC futures long/short strategies to evaluate the operational effectiveness of forecasting models in real-world financial decision-making.
4. Model Validation: The authors benchmarked the performance of over ten SOTA time series forecasting models using the proposed datasets, metrics, and tasks.

**Dataset Code Accessibility:**

Yes

**Dataset Code Comments:**

Both the dataset and code are provided and readable.

**Ethical Considerations:**

No, there are no or only very minor ethics concerns

**Final Justification:**

I would like to thank the authors for their responses to my review comments. I have also reviewed the authors' responses to all reviewers and believe that the added comparisons and discussions during the rebuttal will make the paper more impactful. Therefore, I will increase my score to 5.

**Limitations Weaknesses:**

1. Insufficient Integration of Appendix Information:

a) Key details about the datasets (e.g., data sources, collection processes) are only mentioned in the appendix. Although these details are crucial for understanding the reliability and representativeness of the datasets, they are not even mentioned in the main paper. For instance, Section 3.1 describes the datasets but omits their origins and methods of collection, which are instead buried in Appendix A.
b) Similarly, the limitations of the work are discussed in Appendix E but are absent from the main paper.

A brief mention or redirection to these sections within the main text would improve accessibility and transparency.

**Strengths Contributions:**

1. Clarity and Organization: The paper is well-structured and uses figures effectively to explain the framework, contributions, and experimental results. The main components of the framework are clearly delineated.
2. Extensive Experiments: The paper validates the proposed framework by evaluating more than ten time series forecasting models. This extensive experimentation adds credibility to the framework and provides a valuable reference for future research.
3. Detailed Appendix: While relegated to supplementary material, the appendix contains extensive details about datasets, experimental setups, and discussions. This enhances the reproducibility and transparency of the work.

---

> ### Author Rebuttal · Authors · 2025-07-31
>
> First of all, we would like to thank you for taking the time to carefully review our paper. We believe that your suggestion of briefly mentioning the key details of the dataset and research limitations from the appendix in the main text will enhance accessibility and transparency. We will follow these suggestions in the revision.
>
> We will now address your concerns and questions one by one:
>
> **W1: Insufficient integration of appendix information**
>
> a. We agree with this point. Elaborating on key details in the main text will help readers understand the reliability and representativeness of the dataset. We will incorporate these details into the dataset section of the main text in the revised version. Thank you for pointing this out.
>
> b. We also agree with the suggestion to adjust the content related to limitations. We will mention them in the discussion or conclusion section of the main text in the revised version. We concur that briefly referencing or signposting these sections in the main text will enhance accessibility and transparency.
>
> We would like to sincerely thank you again for your review and specific suggestions, which enable us to further improve our paper.

---

> > ### Comment · Reviewer_CYiu · 2025-08-04
> >
> > I would like to thank the authors for their responses to my review comments. I have also reviewed the authors' responses to all reviewers and believe that the added comparisons and discussions during the rebuttal will make the paper more impactful. Therefore, I will increase my score to 5.

---

> > > ### Author Response · Authors · 2025-08-05
> > >
> > > Thank you sincerely for your feedback and for reconsidering the assessment of our paper based on the discussions among all reviewers during the rebuttal! This has greatly encouraged us and motivated us to further refine and complete this work, so as to make more meaningful contributions to the time series forecasting community.

---

### Note · Authors · 2025-08-13

**Dear Senior Area Chairs, Area Chairs, and Reviewers**

We sincerely thank all reviewers, area chairs, and senior area chairs for your active participation and constructive feedback, which have greatly enhanced our work. During the rebuttal phase, we have addressed all core concerns raised by the reviewers, and the reviewers have explicitly expressed their acceptance of this work. Finally, we summarize the main content as follows.

**Key concerns addressed:**

- **Covering foundation model-based comparison benchmarks.** Following the reviewers' suggestions, we have added time series foundation models as comparison benchmarks and conducted experiments on multivariate-to-multivariate prediction tasks as well as multivariate-to-univariate prediction tasks. By comparing with small models, we find that after fine-tuning, large models exhibit more competitive performance on these financial time series datasets that have not been released before.

- **Adding additional reviews and comparisons.** Following the reviewers' suggestions, we have expanded the review of previous work on financial prediction and further clarified the differences between our work and prior studies in terms of prediction horizons, evaluation methods, model selection, and types of datasets. We have also added discussions on factors that cannot be ignored in real-world scenarios, including transaction costs, investment experiment design, analysis of evaluation metrics, and visual analytics.

- **Adjustments to the paper structure.** Following the reviewers' suggestions, we also relocate critical technical details and key results from the appendix to relevant sections within the main text, and add redirects to strengthen content relevance.

**Revision Integration:**

All suggestions from reviewers in the rebuttal, including adding comparisons with large model benchmarks, expanding reviews of related work in the field, adjusting the paper structure, and analyzing key details, will be integrated into the final revised version. This enhances the clarity and rigor of our work, and we will further expand its impact by continuously supplementing more new benchmarks and high-quality datasets.

FinTSBridge bridges the gap between advanced general time-series prediction models and practical financial time-series scenarios. We expect this work can make meaningful contributions to the time-series prediction community.

Thank you again for your invaluable guidance.

Sincerely,

The Authors

---

### Decision · Program_Chairs · 2025-09-18

**Decision:**

Reject

**Comment:**

This submission digs into financial time series data for asset-pricing. This is, of course, an old field with many distinct problems with underlying financial commonalities.  The reviewers raised a number of substantive issues that the authors agree they would like to work on. The work as-is is promising, and we hope the authors are correct that they can address the interesting concerns raised, but the paper will need to be re-reviewed and hopefully will have shown that it can be substantially clearer and better placed within the context of the state of the art. We sincerely hope the authors find the reviews constructive and helpful as they polish this work and make it hopefully more impactful.

The reviewers recommended acceptance, but NeurIPS must maintain high standards and is constrained in how many submissions can be accepted. The area chairs feel that these reviewers were somewhat more generous in their ratings than other reviewers for other papers.